# Coding and non-coding roles of MOCCI (C15ORF48) coordinate to regulate host inflammation and immunity

Cheryl Q. E. Lee[1], Baptiste Kerouanton[2], Sonia Chothani[2], Shan Zhang[2], Ying Chen[3], Chinmay Kumar Mantri[4], Daniella Helena Hock[5], Radiance Lim[2], Rhea Nadkarni[2], Vinh Thang Huynh[3,6], Daryl Lim[7], Wei Leong Chew[7], Franklin L. Zhong[8,9], David Arthur Stroud[5], Sebastian Schafer[2,10], Vinay Tergaonkar[3,11], Ashley L. St John[4,12,13], Owen J. L. Rackham[2] & Lena Ho[1,2,3✉]

Mito-SEPs are small open reading frame-encoded peptides that localize to the mitochondria to regulate metabolism. Motivated by an intriguing negative association between mito-SEPs and inflammation, here we screen for mito-SEPs that modify inflammatory outcomes and report a mito-SEP named "Modulator of cytochrome C oxidase during Inflammation" (MOCCI) that is upregulated during inflammation and infection to promote host-protective resolution. MOCCI, a paralog of the NDUFA4 subunit of cytochrome C oxidase (Complex IV), replaces NDUFA4 in Complex IV during inflammation to lower mitochondrial membrane potential and reduce ROS production, leading to cyto-protection and dampened immune response. The *MOCCI* transcript also generates miR-147b, which targets the *NDUFA4* mRNA with similar immune dampening effects as MOCCI, but simultaneously enhances RIG-I/MDA-5-mediated viral immunity. Our work uncovers a dual-component pleiotropic regulation of host inflammation and immunity by MOCCI (C15ORF48) for safeguarding the host during infection and inflammation.

[1] Institute of Medical Biology, A*STAR, Singapore, Singapore. [2] Duke-NUS Medical School, Program in Cardiovascular and Metabolic Disorders, Singapore, Singapore. [3] Institute of Molecular and Cell Biology, A*STAR, Singapore, Singapore. [4] Duke-NUS Medical School, Program in Emerging Infectious Diseases, Singapore, Singapore. [5] Department of Biochemistry and Molecular Biology, The Bio21 Molecular Science & Biotechnology Institute, University of Melbourne, Melbourne, VIC, Australia. [6] Lee Kong Chian School of Medicine, Nanyang Technological University Singapore, Singapore, Singapore. [7] Genome Institute Singapore, A*STAR, Singapore, Singapore. [8] Nanyang Technological University, Skin Diseases and Wound Repair Program, Singapore, Singapore. [9] Skin Research Institute of Singapore, A*STAR, Singapore, Singapore. [10] National Heart Research Institute Singapore, National Heart Centre Singapore, Singapore, Singapore. [11] Department of Pathology, Yong Loo Lin School of Medicine, National University of Singapore, Singapore, Singapore. [12] Department of Microbiology and Immunology, Yong Loo Lin School of Medicine, National University of Singapore, Singapore, Singapore. [13] Department of Pathology, Duke University Medical Center, Durham, NC, USA. ✉email: lena@ho-lab.org

Activation of the endothelium by inflammatory triggers initiates a cascade of signaling pathways with the aim of danger signal clearance and resolution of the inflammatory response, ultimately returning the endothelium to its basal state[1]. Failure to do so results in excessive vascular inflammation that leads to tissue damage in acute settings, and cardiovascular diseases such as atherosclerosis and fibrosis in chronic settings. Although broad immunosuppressive strategies such as tumor necrosis factor-alpha (α), interleukin-1beta (IL-1β), or interleukin-6 (IL-6) blockade with neutralizing antibodies have yielded promising results in clinical trials to reduce vascular inflammation[2], their considerable side effects and limited efficacy indicate a need for alternative, more specific targets, potentially downstream of these cytokines, and preferably specific to the endothelium.

Mitochondria are multifaceted organelles most well-known for their roles in cellular energy production via oxidative phosphorylation and in apoptosis via cytochrome c release. Besides energetic homeostasis, the role of mitochondria in regulating the onset and outcomes of inflammation is becoming increasingly appreciated[3]. A classic example is the production of reactive oxygen species (ROS) from the respiratory chain. Once thought to be purely pathogenic, it is now appreciated that low or physiological levels of ROS play normative roles in both pro-inflammatory and pro-resolving signaling pathways, while excessive ROS can lead to chronic activation of pro-inflammatory mechanisms that cause tissue damage[4]. The mitochondria are also known to be a proximal site for the activation of interferon response (IR) via the localization of MAVS[5] or the activation of inflammasome via the extrusion of cardiolipin[6]. Leakage of mitochondrial DNA can also trigger IR or inflammatory cell death through the c-GAS/STING pathway[7]. Furthermore, metabolites of the Krebs cycle such as succinate and itaconate are increasingly recognized as modulators of inflammation; for example, through the regulation of the NLRP3 inflammasome and the IR pathway[8,9]. Although mitochondrial dysfunction is widely implicated in the pathogenesis of cardiovascular diseases, the specific function of mitochondria in endothelial health and dysfunction has received relatively little attention. In particular, how endothelial mitochondria respond to and modulate an inflammatory response is poorly understood.

Previously, our lab reported a preponderance of small open reading frame (sORF)-encoded peptides, defined as proteins smaller than 100 residues that localize to the mitochondria (mito-SEPs)[10]. These peptides have functions in diverse processes in the mitochondria, including electron transport, lipid metabolism, and calcium homeostasis. We asked the question of whether mito-SEPs might play hitherto unappreciated roles in regulating inflammation. For instance, Fitzgerald and colleagues recently identified a mitochondrial peptide Mm47 that is required for the activation of the NLRP3 inflammasome[11].

In this study, we perform a proteogenomic screen to find mito-SEPs in primary human aortic endothelial cells (HAECs) that can promote the resolution of inflammation. We report the discovery of modulator of cytochrome C oxidase during Inflammation (MOCCI), a mito-SEP encoded by *C15ORF48*. In concert with miR-147b in the 3′ untranslated region (UTR) of *C15ORF48*, MOCCI replaces its paralog NDUFA4 in Complex IV (CIV). This dampens CIV activity and protects host tissue against excessive immune pathology by reducing cell death and cytokine production. Furthermore, miR-147b exerts potent antiviral effects by enhancing the IR through the RIG-I/MDA5 pathway. Altogether, the coding and non-coding functions of the *C15ORF48* synergize to protect the host during infection, illustrating how small peptides coordinate with their encoding transcripts to achieve maximal biological impact.

## Results

**Proteogenomic screen of inflammation-associated Mito-SEPS (*i*-Mito-SEPs) identifies MOCCI, encoded by *C15ORF48*.** Mito-SEPs are peptides encoded from sORFs in the nuclear genome that are imported into the mitochondria for various facets of mitochondrial function. While characterizing mito-SEPS involved in metabolic regulation, we uncovered an unexpected negative association between mito-SEPS and inflammatory pathways such as interferon signaling in transcriptomes of human hearts with dilated cardiomyopathy[12], a setting of chronic vascular inflammation (Fig. 1a). This motivated us to devise a strategy to uncover hitherto uncharacterized mito-SEPs that are involved in regulating inflammation, specifically in the vascular system. To this end, we adopted an unbiased proteogenomic strategy based on ribosome-profiling (Ribo-seq) coupled with RNA sequencing (RNA-seq) of primary HAECs treated with IL-1β (Fig. 1b). We chose 45 min, 12 and 24 h post-treatment to capture the full course of the inflammatory response including resolution, as indicated by the downward trend of activated pro-inflammatory markers NF-κB activity, ICAM-1 and VCAM-1 (Fig. 1b). We used HAECs from two healthy male donors in triplicate to avoid donor-specific effects and performed paired Ribo-seq with RNA-seq. Our Ribo-seq protocol generated 28–30 nucleotide ribosome protected fragments (RPFs) that displayed an average in-frame periodicity of 81.98% across annotated protein coding ORFs (Supplementary Fig. 1a, b). We then used RiboTaper[13] to identify all expressed ORFs from our dataset that belong to known consensus coding sequence (CCDS) regions of the genome. These were filtered to include only ORFs encoding peptides less than 100 amino acids and exclude those overlapping with larger proteins (Fig. 1c). Only those that meet the quality control metrics based on the parameters of known SEPs were retained (see methods). Of these, we filtered for those with significant differential expression at the RNA level during IL-1β stimulation (Supplementary Data 1). Lastly, candidate genes and peptides were passed through our mitochondrial protein motif and gene signature-based mito-SEP prediction pipeline as described previously[10] to identify mitochondrial SEPs (Fig. 1c, d). A candidate was assigned a mitochondrial gene signature if it correlated with MitoCarta genes in two or more of four tissues with pathological or physiological inflammation (Fig. 1d, Table 1, and Supplementary Data 1). We identified 21 putative inflammatory mito-SEPS (*i*-Mito-SEP) (Fig. 1c). Finally, we triaged each surviving candidate according to its degree of differential expression and whether it was positive for mitochondrial motif, gene signature, or both (Fig. 1e). The top *i*-Mito-SEP candidate was an 83-aa peptide encoded by the *C15ORF48* gene, also called *NMES-1*, which showed the largest upregulation following IL-1β treatment and which was predicted to be mitochondrial by both prediction methods (Fig. 1e). We also confirmed that Ribo-seq reads were increased upon IL-1β treatment, indicating that there is more translation during inflammation (Fig. 1f). In-frame P-site reads were also detected across the entire ORF (Fig. 1g). This allowed us to discriminate between alternate translation initiation sites and accurately determine the start site of the SEP.

Gene module association determination (G-MAD) uses a large cohort of expression data to predict function of a gene based on its expression pattern with other genes. In human tissues, *C15ORF48* expression is correlated to genes involved in the regulation of cytokine activity, in particular IL-10; response to microbial infection, interferon signaling, and leukocyte chemotaxis (Supplementary Fig. 1c). Notably, in some tissues such as the artery, *C15ORF48* is negatively associated with genes involved in cellular and viral mRNA translation (Supplementary Fig. 1c). These results suggest that *C15ORF48* might play a role in regulating the innate immune response against pathogens such as

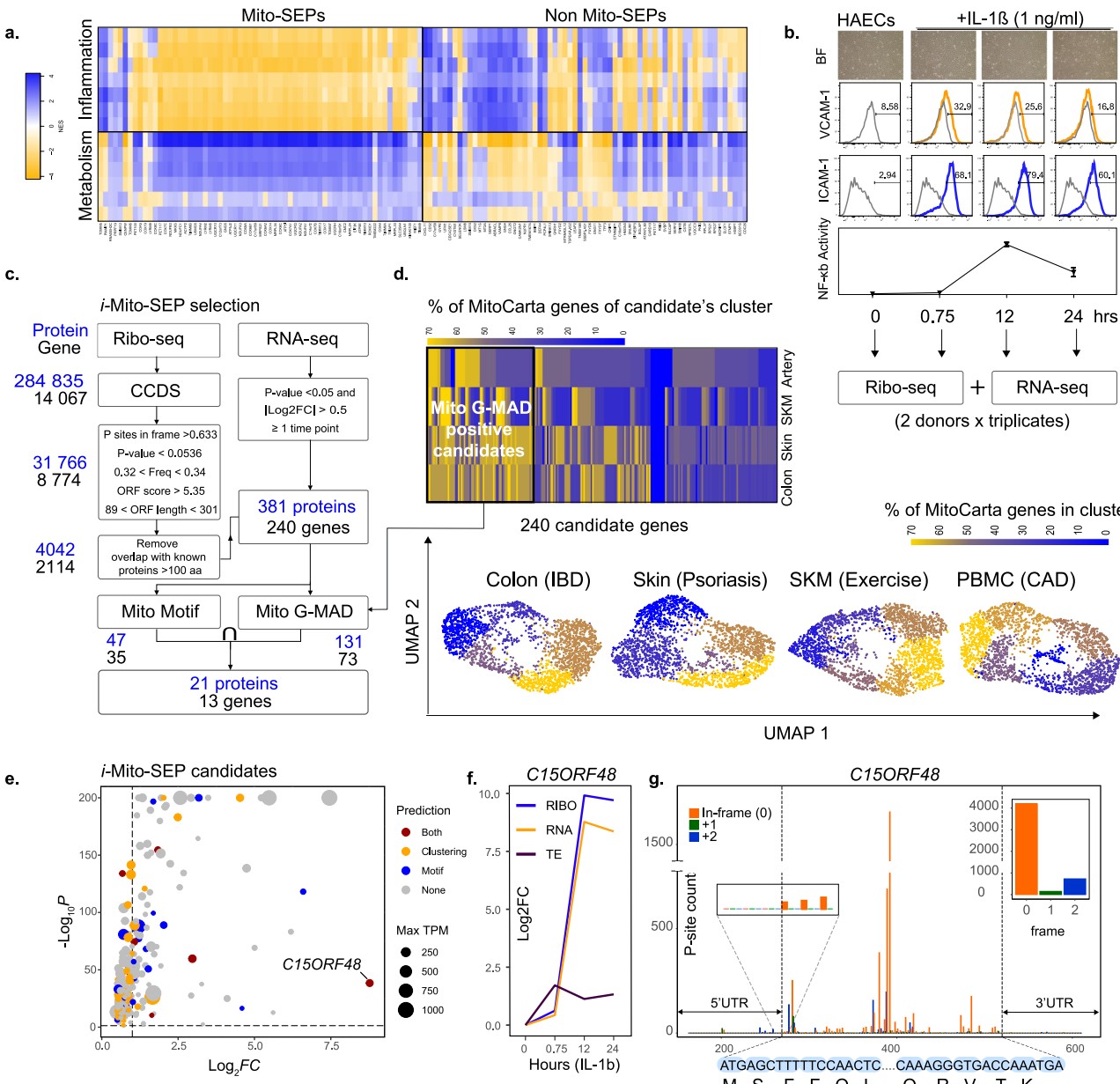

**Fig. 1 Proteogenomic screen in human endothelial cells (HAECs) identifies MOCCI as an inflammatory Mito-SEP (i-Mito-SEP). a** WGCNA-GSEA analysis of mito versus non-mito-SEPs in human failing heart reveals strong anti-correlation of mito-SEPs with inflammation. Each row represents one pathway and the pathways are separated into those that are involved in metabolism and those that are involved in inflammation. Color scale = NES score. Dataset used: European Genome-Phenome Archive EGAS00001002454[12]. **b** HAECs from two healthy male donors were treated with 1 ng/mL IL-1β for 45 min, 12 and 24 h. All timepoints including untreated controls were analyzed by RNA-seq and Ribo-seq in triplicate. NF-kb reporter activity, ICAM-1, and VCAM-1 expression illustrate the dynamics of the HAEC response: an increase in inflammation up to 12 h followed by resolution thereafter. Data for NF-kb reporter activity are presented as mean ± SEM. $n = 4$ biological replicates for NF-kb reporter activity. **c** Workflow to identify i-Mito-SEPs. Blue numbers indicate the number of SEPs called by RiboTaper and black numbers indicate the number of genes encoding those SEPs (see methods). **d** Summary of the mitochondrial gene signature workflow. (Bottom) UMAP of gene module association determination (G-MAD) scores of MitoCarta and random genes in human colon, skin, skeletal muscle (SKM), and PBMCs undergoing inflammation. Color scale indicates the percentage of mitochondrial genes in each cluster. Based on their G-MAD, the 240 candidate genes from **c** were assigned to a cluster in each dataset. (Top) Heatmap depicting the percentage of mitochondrial genes in a candidate's cluster in each of the four datasets. Each column represents one candidate. Top scoring candidates were returned to the pipeline in **c** for further consideration. Datasets used were: GSE11223[84], GSE14905[83], GSE120862[82], and GSE9820[81]. **e** Volcano plot of the maximal fold change of the 240 candidate genes during the IL-1β treatment compared to untreated cells. The size of each dot represent the highest TPM reached at any timepoint, while its color indicates if candidate SEP was predicted to localize to the mitochondria by motif, G-MAD or both. The p adjusted values were attained by applying the Wald chi-squared test, followed by correction for multiple testing using the Benjamini and Hochberg method through the DESeq2 pipeline. **f** MOCCI transcript and translation levels increase by 1000-fold after IL-1β treatment. TE translational efficiency. **g** Ribo-seq derived p-site read coverage across the MOCCI locus. In-frame (0) and out-of-frame (+1,+2) colored in orange, green, and blue respectively. Inset on the left shows that p-site reads are detected only at the start codon. Inset on the right summarizes the total number of in-frame (0) and out-of frame (+1,+2) reads in dataset.

**Table 1 List of published datasets that were used.**

| Tissue | Condition | Organism | Data type | Database | GSE number |
|---|---|---|---|---|---|
| PBMC | CAD/LPS | Human | Bulk RNA-seq | GEO | GSE9820[81] |
| SKM | Exercise | Human | Bulk RNA-seq | GEO | GSE120862[82] |
| Skin | Psoriasis | Human | Bulk RNA-seq | GEO | GSE14905[83] |
| Colon | IBD | Human | Bulk RNA-seq | GEO | GSE11223[84] |
| Skin | Inflammasome activation | Human | Bulk RNA-seq | GEO | GSE85791[117] |
| Artery | CAD model | Mouse | scRNA-seq | GEO | GSE131776[94] |
| PBMC | IFNb treatment | Human | scRNA-seq | GEO | GSE96583[118] |
| Lung | Influenza infection | Mouse | scRNA-seq | GEO | GSE107947[96] |
| Dendritic cells | Chlamydia infection | Human | Microarray | GEO | GSE12806[119] |
| Dendritic cells | Hypoxia | Human | Microarray | GEO | GSE6863[120] |
| Bronchial epithelium | Influenza infection | Human | Microarray | GEO | GSE48466[121] |
| Heart | Dilated cardiomyopathy | Human | Bulk RNA-seq | European Genome-phenome Archive | EGAS00001002454[12] |

bacteria and viruses. Indeed, single-cell RNA-seq (scRNA-seq) data of human peripheral blood mononuclear cells (PBMCs) show that *C15ORF48* mRNA is highly expressed in monocytes at basal and interferon-stimulated state (Supplementary Fig. 2a). In non-immune cells however, *C15ORF48* expression is low in healthy tissues, and upregulated in endothelial cells in conditions of chronic inflammation such as atherosclerosis (Supplementary Fig. 2b). Similarly, during influenza infection in mice, *C15ORF48* is upregulated in granulocytes and monocyte precursors as well as lymphatic endothelial cells (Supplementary Fig. 2c). These data together point to a constitutive role of *C15ORF48* in monocytes and an inducible function in endothelial and phagocytes cells during both acute pathogen-induced and chronic sterile inflammation.

**MOCCI is a paralog of NDUFA4, 14th subunit of Complex IV.** Consistent with our finding that *C15ORF48* encodes a putative mito-SEP, G-MAD shows strong and positive association across all tissues between *C15ORF48* and genes encoding respiratory chain CIV (red), in addition to inflammation (Supplementary Fig. 1c). The SEP encoded by *C15ORF48*—which we refer to hereafter as MOCCI–has previously been implicated in a large interactome study to interact with components of the electron transport chain (ETC), where it was overexpressed as a FLAG-tagged bait to capture interactors, although no definitive function was assigned to the protein[14]. MOCCI is conserved evolutionarily across vertebrates (Fig. 2a) but not present in fungi. MOCCI contains a sole B12D domain (NADH-ubiquinone reductase complex 1 MLRQ subunit) in an architecture that is found also in NDUFA4 and NDUFA4L2 (Fig. 2a, b and Supplementary Fig. 2d). Indeed, MOCCI shares a strong conservation at the amino acid level with NDUFA4/NDUFA4L2 proteins, which are also SEPs by definition (Supplementary Fig. 2d). NDUFA4 was originally assigned to be an accessory unit of Complex I (CI) (NADH:ubiquinone oxidoreductase) but was later found to be the 14th subunit of CIV[15–17], which we confirmed by co-migration between NDUFA4 and COX4 using blue native polyacrylamide gel electrophoresis (BN-PAGE) of digitonin-solubilized mouse heart mitochondria (Supplementary Fig. 2e). To elucidate the function of MOCCI and verify the observed association with CIV, we first determined its subcellular localization in MOCCI-overexpressing HEK293T cells where MOCCI is detected by western blot in mitochondria-enriched fractions (Fig. 2c). Differential extraction assays of isolated mitochondria indicate that MOCCI is localized to the inner mitochondrial membrane (IMM) (Fig. 2d), consistent with the presence of a predicted trans-membrane domain, and confirmed by Proteinase K (PK) protection assays (Fig. 2e). To verify that MOCCI is part of the ETC, specifically CIV, we overexpressed the ORF encoding mouse

MOCCI[FLAG] by AAV9-mediated (AAV-MOCCI) delivery to the heart, an endothelial rich organ, and performed BN-PAGE analysis of isolated tissue mitochondria (Fig. 2f). Western blotting with anti-MOCCI antibody demonstrated that the MOCCI[FLAG] co-migrated with MTCO-1, the catalytic subunit of CIV (Fig. 2f). This was confirmed using an anti-FLAG antibody (Supplementary Fig. 2f). Furthermore, using two-dimensional BN-SDS PAGE, we confirmed the complete colocalization of MOCCI with MTCO-1, with the highest occupancy in CIV monomers and dimers, with some comigrating in supercomplexes (Fig. 2g).

**CIV Composition Switches from NDUFA4-to-MOCCI during Inflammation.** In addition to the *MOCCI* ORF, the 3′UTR of *C15ORF48* hosts *hsa-miR-147b* (Fig. 2a), an evolutionarily conserved miRNA that has previously been implicated to target *NDUFA4* mRNA via a conserved seed sequence in its 3′UTR (Fig. 2b). In HEK293T cells, transfection of miR-147b mimic downregulated *NDUFA4* mRNA and NDUFA4 protein (Supplementary Fig. 3a, b). These data suggest that upregulation of the *C15ORF48* mRNA has two outcomes: incorporation of MOCCI into CIV and miR-147b-mediated downregulation of the MOCCI paralog NDUFA4, essentially effecting a subunit switch in CIV. Since translation of an mRNA versus its processing into a pre-miRNA are mutually exclusive events, we first investigated if both processes were occurring in HAECs. In line with the HAEC Ribo-seq data, the MOCCI peptide is translated in about 25.7% of HAECs in response to IL-1β (Fig. 3a and Supplementary Fig. 3c), where it shows clear localization to the mitochondria as judged by co-localization with TOMM20 (Fig. 3b). At the same time, IL-1β treatment in HAECs robustly induced the biogenesis of mature miR-147b (Fig. 3c). This suggests that upon IL-1β stimulation, HAECs generate both MOCCI and miR-147b, and together these might function synergistically to enforce the switch in CIV from NDUFA4-to- MOCCI. Importantly, HAECs demonstrated NDUFA4-to-MOCCI switch mediated by upregulation of endogenous *C15ORF48* as visualized by intracellular flow cytometry in response to IL-1β (Fig. 3d). We validated this NDUFA4-to-MOCCI switch in A549 lung epithelial cells (Fig. 3e), which like HAECs upregulate both *C15ORF48* transcript and *miR-147b* in response to IL-1β treatment (Supplementary Fig. 3d). Conversely, CRISPR/Cas9-mediated disruption of the MOCCI ORF within *C15ORF48* ("MOCCI-KO", Supplementary Fig. 3e) reduced MOCCI upregulation (Supplementary Fig. 3f) and NDUFA4 downregulation (Supplementary Fig. 3g) in HAECs treated with IL-1β. However, miR-147b production was also disrupted in the KO cells, despite normal levels of mutant *C15ORF48* mRNA (Supplementary Fig. 3h). We were therefore unable to separate the effects of MOCCI and miR-147b using MOCCI-KO HAECs. To further clarify the relative contributions of the MOCCI

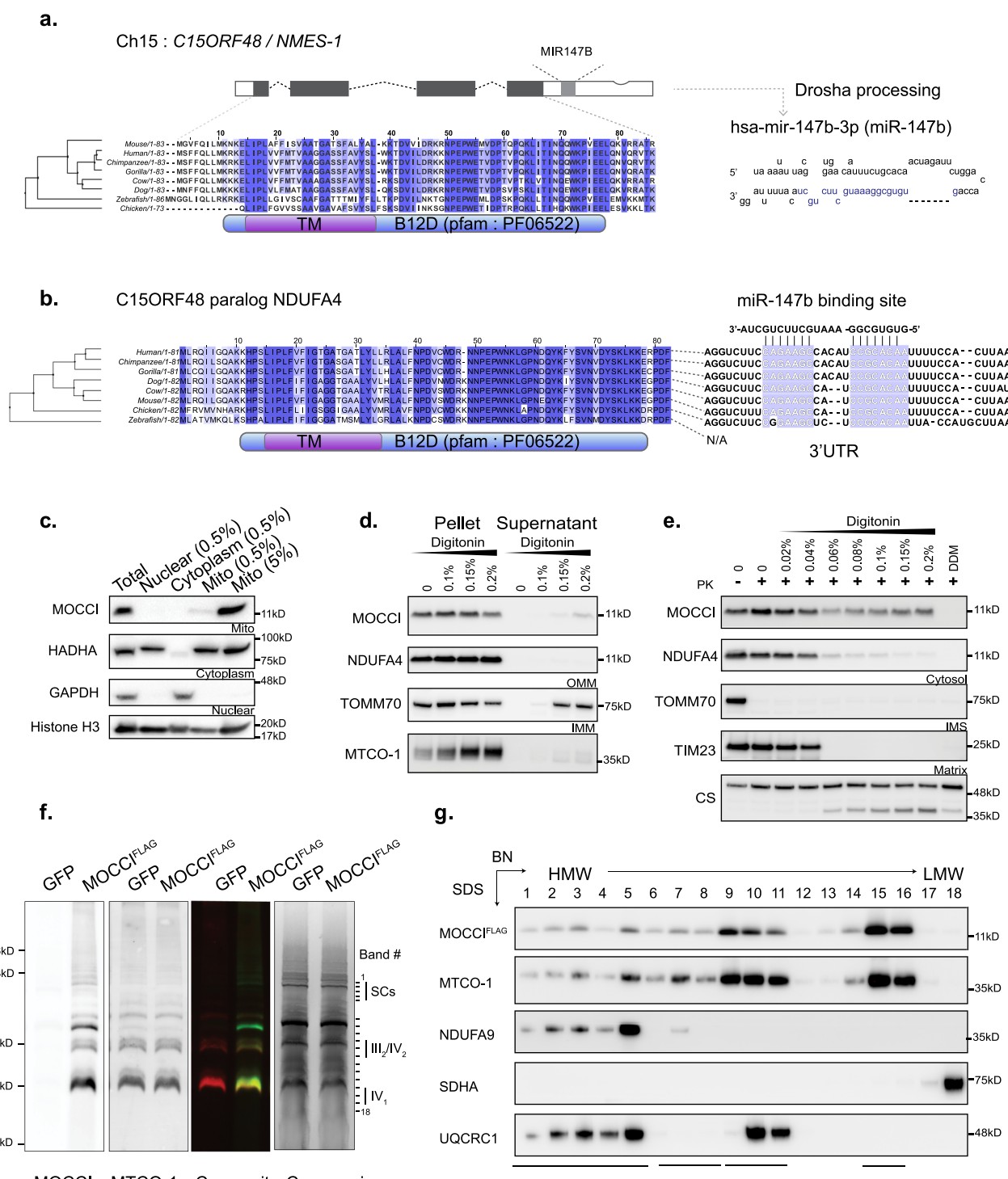

peptide and miR-147b to the observed NDUFA4-to-MOCCI switch, we stably expressed the ORF of human MOCCI (hereafter referred to as "MOCCI") and full length *C15ORF48* mRNA (which produces both MOCCI and miR-147b and is hereafter referred to as "WT-mRNA") via lentiviral transduction. Consistent with the findings from MOCCI-KO HAECs, we found that MOCCI alone was sufficient to lower NDUFA4 protein levels by 61.2%, while WT-mRNA reduced NDUFA4 protein levels by 80.6%, suggesting that MOCCI is equally, if not more effective than miR-147b at NDUFA4 attenuation (Fig. 3f and Supplementary Fig. 3i). Indeed, mutating the start codon of the MOCCI ORF within *C15ORF48* mRNA to isoleucine (referred to hereafter

as "ATGmut-mRNA"), while preserving miR-147b biogenesis (Fig. 3g), led to only 31.5% downregulation of NDUFA4 protein (Fig. 3f and Supplementary Fig. 3i). This was also validated by western blot, which showed NDUFA4 protein levels to be much lower in MOCCI and WT-mRNA HAECs while ATGmut HAECs showed only intermediate downregulation (Fig. 3h). While miR-147b from WT and ATGmut-mRNA expression reduced *NDUFA4* mRNA by about 50%, MOCCI did not affect *NDUFA4* mRNA levels (Supplementary Fig. 3j). These results together argue that MOCCI alone is sufficient to downregulate NDUFA4 protein level without affecting its mRNA levels. MiR-147b contributes by attenuating *NDUFA4* mRNA and potentially

**Fig. 2 MOCCI is a subunit of Complex IV (cytochrome C oxidase). a** The *MOCCI* ORF is conserved across vertebrates and contains a predicted transmembrane (TM) domain and the B12D domain found only in NADH-ubiquinone reductase complex 1 MLRQ subunits. The *MOCCI* transcript is also the host gene for hsa-miR-147b (right), located in the 3′UTR of the transcript. The letters in blue are the sequence of the mature miRNA. **b** MOCCI is a paralog of NDUFA4 which also contains both the B12D and TM domains. The 3′UTR of the *NDUFA4* transcript contains top target site of miR-147b-3p as predicted by TargetScan[114], miRDB[115], and miRBase[116]. It is not clear if the binding site in chicken *Ndufa4* is functional, as it is not as well conserved. **c** MOCCI was found in mitochondria-enriched fractions of HEK293T transfected with MOCCI ORF-expressing plasmid (MOCCI-HEK293T). Values given indicates percentage of the total fraction loaded onto the gel. $n = 2$ biological replicates. Source data are provided as a Source Data file. **d** Differential solubilization of MOCCI-HEK293T mitochondria indicated that MOCCI was located in the inner membrane (IMM). TOMM70 and MTCO-1 serve as control for outer (OMM) and inner (IMM) mitochondrial membrane proteins, respectively. $n = 1$ biological replicate. Source data are provided as a Source Data file. **e** Proteinase K protection assay of MOCCI-HEK293T mitochondria further confirmed IMM localization of MOCCI. $n = 1$ biological replicate. Source data are provided as a Source Data file. **f** Blue native PAGE (BN-PAGE) of mouse heart mitochondria with AAV9-mediated MOCCI-FLAG expression (AAV-MOCCI). Probing with α-MOCCI and α-MTCO-1 antibodies, the core enzymatic subunit of CIV, reveal co-migration of MOCCI signal with CIV. AAV-GFP was used as a negative control. Coomassie blue stain of the gel is shown on the right. $n = 4$ biological replicates. **g** Two-dimensional BN-SDS-PAGE to confirm co-migration of MOCCI and CIV. The lanes correspond to the bands annotated on the Coomassie blue-stained gel in (**f**). $n = 2$ biological replicates. Source data are provided as a Source Data file.

synergizes with MOCCI to reduce NDUFA4 protein. Altogether, MOCCI and miR-147b exert different levels of control on NDUFA4 protein and mRNA levels (Supplementary Fig. 3k).

To confirm these in vitro observations, we analyzed the proteome of mitochondria isolated from AAV-MOCCI and AAV-GFP injected mice by quantitative mass spectrometry, and found that MOCCI was sufficient to downregulate NDUFA4 protein levels in vivo (Fig. 4a and Supplementary Data 2). We confirmed by PAGE analysis that AAV-MOCCI mitochondria had a 60% reduction in both total (SDS-PAGE) and CIV-associated (BN- and BN-SDS PAGE) NDUFA4 (Fig. 4b, c and Supplementary Fig. 4a). Levels of CIV (detected by COX4) and other complexes were unaltered (Fig. 4a, b and Supplementary Fig. 4b). These results confirm that MOCCI alone is sufficient to downregulate NDUFA4 protein levels, potentially by excluding NDUFA4 from CIV. miR-147b synergizes with MOCCI to mediate the switch by targeting *NDUFA4* mRNA. Since the predicted structure of MOCCI and that of NDUFA4 are superimposable (Fig. 4d), we postulate that they occupy the same position in the CIV structure i.e., at the surface of the complex adjacent to MTCO-1 and in close proximity to the heme-copper binuclear catalytic core (Fig. 4e). To test if MOCCI and NDUFA4 are mutually exclusive in CIV complexes, we immunoprecipitated MOCCI-FLAG under native non-denaturing conditions in both total (Fig. 4f) and sucrose-gradient purified CIV monomeric complexes (Supplementary Fig. 4c) in AAV-MOCCI and AAV-GFP heart mitochondria. Pulldown of MOCCI, which enriched MTCO-1 by 231% failed to coimmunoprecipitate NDUFA4 (Fig. 4f, Supplementary Fig. 4c). These data imply that MOCCI and NDUFA4 are not present in the same CIV complexes and support a model in which NDUFA4 and MOCCI are incorporated into alternate assemblies of CIV, or CIV isozymes, raising the question of the functional significance of MOCCI versus NDUFA4-containing CIV complexes.

**MOCCI reduces CIV activity, mitochondrial membrane potential, and ROS production.** We initially sought to define the molecular function of MOCCI in THP-1 monocytes and their macrophage derivatives, which are most commonly employed to investigate mechanisms of inflammatory regulation. However, despite robust upregulation of the *C15ORF48* transcript and mature miR-147b in M1 macrophages treated with IL-1β (Supplementary Fig. 5a, b), the MOCCI peptide is not translated in THP-1 monocytes or THP-1-derived M1 macrophages as detected by flow cytometry (Supplementary Fig. 5c) and western blotting (Supplementary Fig. 5d). Therefore, in vitro cultured human monocytes and macrophages produce only miR-147b from the *C15ORF48* locus. We therefore carried out all subsequent experiments in HAECs and A549 cells. First, we

examined the effect of the NDUFA4-to-MOCCI switch on CIV activity, which can be modulated by the incorporation of paralogous subunits (see Discussion). Since MOCCI alone is sufficient to recapitulate the switch, we measured oxygen consumption and electron flow through CIV using TMPD as an electron donor with heart mitochondria isolated from AAV-MOCCI mice. This experiment utilizes oxygen consumption by CIV as a readout of flux through CIV while CIII is inhibited by Antimycin A, uncoupled from ATP production by FCCP (Fig. 5a). AAV-MOCCI mitochondria had significantly lower CIV electron flux compared to control mitochondria (Fig. 5b), without a significant change in overall uncoupled rotenone-sensitive flux through CI (Supplementary Fig. 5e) or coupled respiration (Supplementary Fig. 5f). These results suggest that MOCCI incorporation lowers CIV activity without affecting respiratory output, which can be attributed to the known large excess capacity and low flux control coefficient of CIV in both isolated mitochondria[18] and intact tissues[19]. Using permeabilized HAECs and A549, we found that MOCCI overexpression led to a small reduction in CIV activity (Fig. 5c). This again did not result in a reduction in overall coupled respiration in response to pyruvate (Supplementary Fig. 5g) in permeabilized cells and did not impact basal or maximal respiration in intact HAECs and A549 (Supplementary Fig. 5h), indicating that MOCCI expression does not compromise respiratory capacity in spite of the reduction in CIV activity. The attenuated CIV activity in MOCCI A549 cells was confirmed using spectrophotometric assays (RCA), which measures CIV enzymatic activity by kinetic measurements of azide-sensitive cytochrome c reduction[20] (Fig. 5d). The reduction in CIV activity was accompanied by a reduction in membrane potential (ΔΨm), as measured using tetramethylrhodamine ethyl ester (TMRE)[21] and ratiometric JC-1 in both HAECs and A549 (Fig. 5e). Lowering membrane potential is predicted to lower ROS production[22,23]. Indeed, we found that HAECs and A549 expressing MOCCI had lower total ROS as measured by 2′,7′-dichlorodihydrofluorescein (DCF) (Fig. 5f) and Amplex Red (Fig. 5g), and mitochondrial ROS as measured by MitoSOX™ Red (Supplementary Fig. 5i) compared to WT cells.

**miR-147b reduces CIV activity but not membrane potential and ROS production.** Depletion of NDUFA4 has been previously reported to reduce CIV activity[15] and impair respiration. Since miR-147b targets NDUFA4, we asked if miR-147b can recapitulate the effects of MOCCI. Transfection of a miR-147b mimic at similar levels as WT-mRNA cells (Supplementary Fig. 6a) reduced CIV activity to intermediate levels (Fig. 6a, b) without impairing respiration (Supplementary Fig. 6b). In contrast, siRNA-mediated knockdown of NDUFA4 significantly lowered

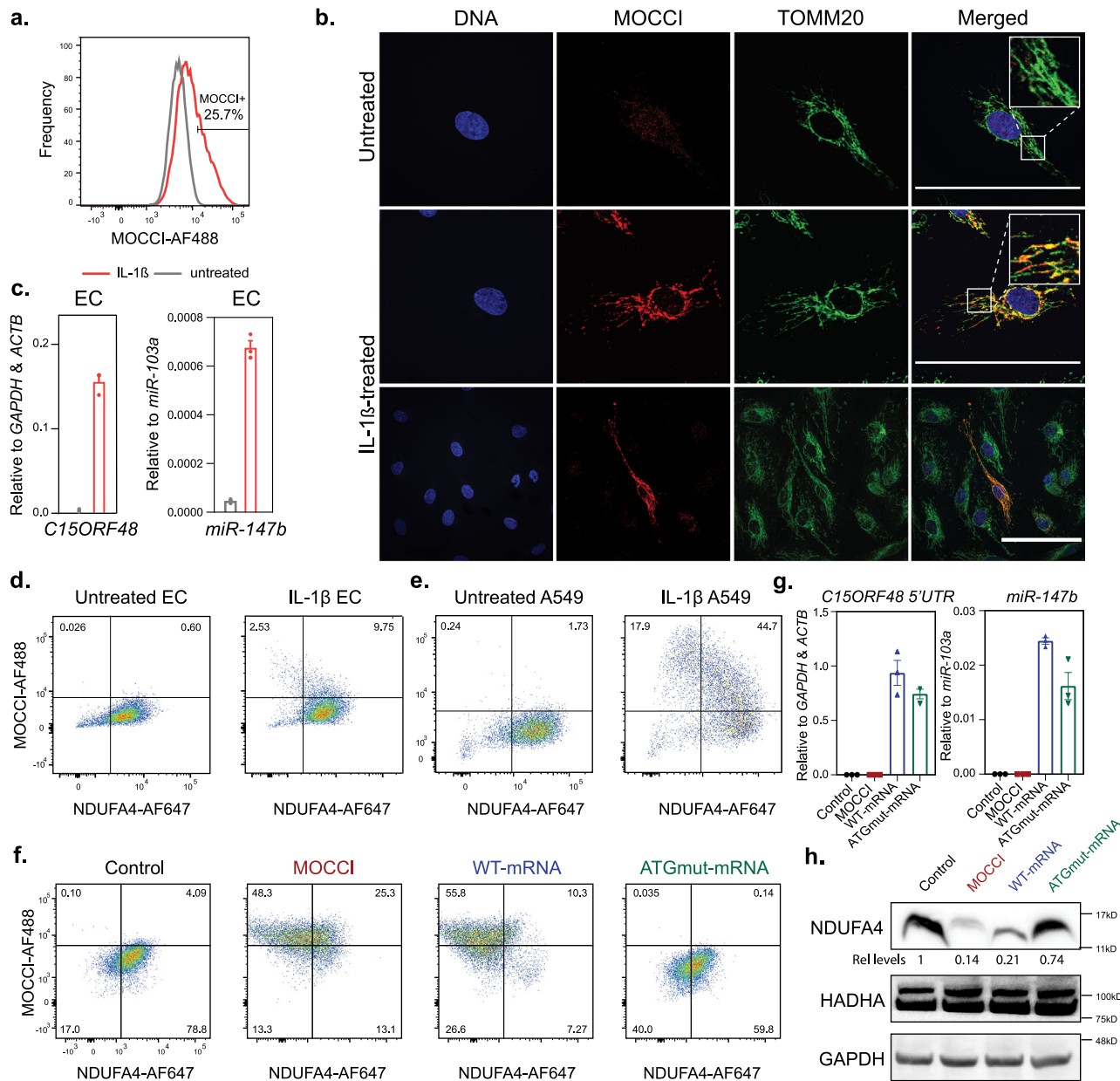

**Fig. 3 MOCCI and *miR-147b* reduce NDUFA4 expression during inflammation. a** Intracellular flow cytometry staining for MOCCI in untreated (gray line) and IL-1β-treated (red line) human aortic endothelial cells (HAECs). Percentage of MOCCI-positive HAECs is 25.7 ± 5.2% (mean ± se), n = 4 biological replicates per condition. **b** Co-immunofluorescent staining for MOCCI and TOMM20, a mitochondrial protein in HAECs. Scale bar = 100 μm, n = 3 biological replicates. **c** *C15ORF48* mRNA and *miR-147b* expression in untreated and IL-1β-treated HAECs. Data are presented as mean ± SEM, n = 3 biological replicates per condition. **d** Intracellular flow cytometry staining for NDUFA4 and MOCCI in untreated or IL-1β-treated HAECs. **e** Intracellular flow cytometry staining for NDUFA4 and MOCCI in untreated or IL-1β-treated A549 cells. **f** Intracellular flow cytometry staining for NDUFA4 and MOCCI in HAECs overexpressing MOCCI, *C15ORF48* WT-mRNA (both MOCCI and miR-147b), and *C15ORF48* ATGmut-mRNA (only miR-147b), n = 3 biological replicates per condition. **g** Levels of *C15ORF48* mRNA (primers binding to the 5'UTR of *C15ORF48*) or miR-147b by qPCR in HAECs with the indicated overexpression. Data were presented as mean ± SEM, n = 3 biological replicates per condition. **h** Western blot of NDUFA4 in HAECs with the indicated overexpression. n = 3 biological replicates. Source data are provided as a Source Data file.

CIV activity to an extent that affected basal respiration in intact cells (Fig. 6a, b and Supplementary Fig. 6b). Similarly, ATGmut-mRNA lowered CIV activity (Fig. 6c) without affecting respiration (Supplementary Fig. 6c). Hence both MOCCI and miR-147b tonically lower CIV activity without affecting the output of the respiratory chain, while complete NDUFA4 removal without replacement by MOCCI (i.e., siNDUFA4) reduces respiration (Supplementary Fig. 6b). Surprisingly, miR-147b did not reduce membrane potential or ROS production in HAECs and A549 (Supplementary Fig. 6d). In line with this, neither WT-mRNA

nor ATGmut-mRNA HAECs or A549 had reduced membrane potential (Fig. 6d) or ROS production (Fig. 6e, f). Rather, expression of WT-mRNA—which produces both MOCCI and miR-147b—completely reversed the drop in membrane potential and ROS production caused by MOCCI (Fig. 6d, f). However, this antagonism is not due to miR-147b within WT and ATGmut-mRNA, since miR-147b transfection did not revert the membrane potential of MOCCI cells back to WT levels (Supplementary Fig. 6d) or increase ROS (Supplementary Fig. 6e), raising the possibility that *C15ORF48* transcript might have additional

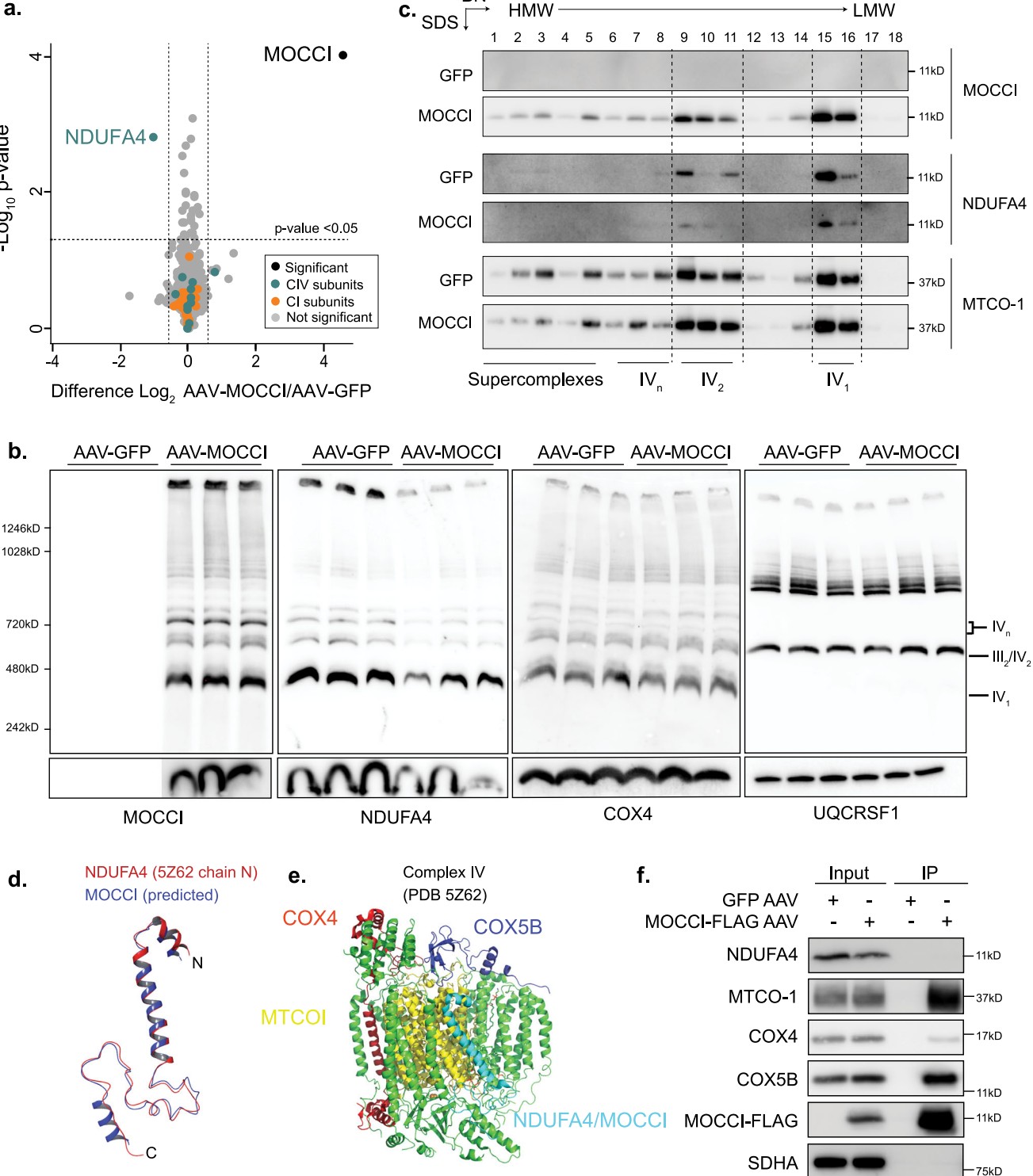

**Fig. 4 NDUFA4 is exchanged for MOCCI in Complex IV during inflammation. a** Volcano plot of proteins detected by quantitative TMT-mass spec in AAV-GFP or AAV-MOCCI mouse heart mitochondria. $p =$ two-tailed Student's $t$ test, $n = 3$ mice per AAV. **b** BN-PAGE (top) and SDS-PAGE (bottom) of AAV-MOCCI/GFP mouse heart mitochondria probed for the indicated respiratory chain proteins. Each lane represents mitochondria from one mouse. Quantification of NDUFA4 shown in Supplementary Fig. 4a. **c** Two-dimensional BN-SDS-PAGE to show downregulation of NDUFA4 in CIV complexes. $n = 3$ biological replicates. Source data are provided as a Source Data file. **d** Alignment of the predicted secondary structure of MOCCI (iTASSER model) and known structure of NDUFA4 in Maestro (Schrodinger). **e** The position of NDUFA4 (ribbon) in CIV cryo-EM structure (PDB 5Z62). All other subunits are depicted in tube representation. MTCO-1 (purple) and the redox centers (HEME-A and $Cu^{2+}$) are highlighted to illustrate the relative position of NDUFA4. **f** Co-IP of NDUFA4 and MOCCI. Digitonin (1%)-solubilized isolated mouse heart mitochondria were immunoprecipitated with anti-FLAG and probed for the proteins as indicated. AAV-GFP served as negative control. Source data are provided as a Source Data file.

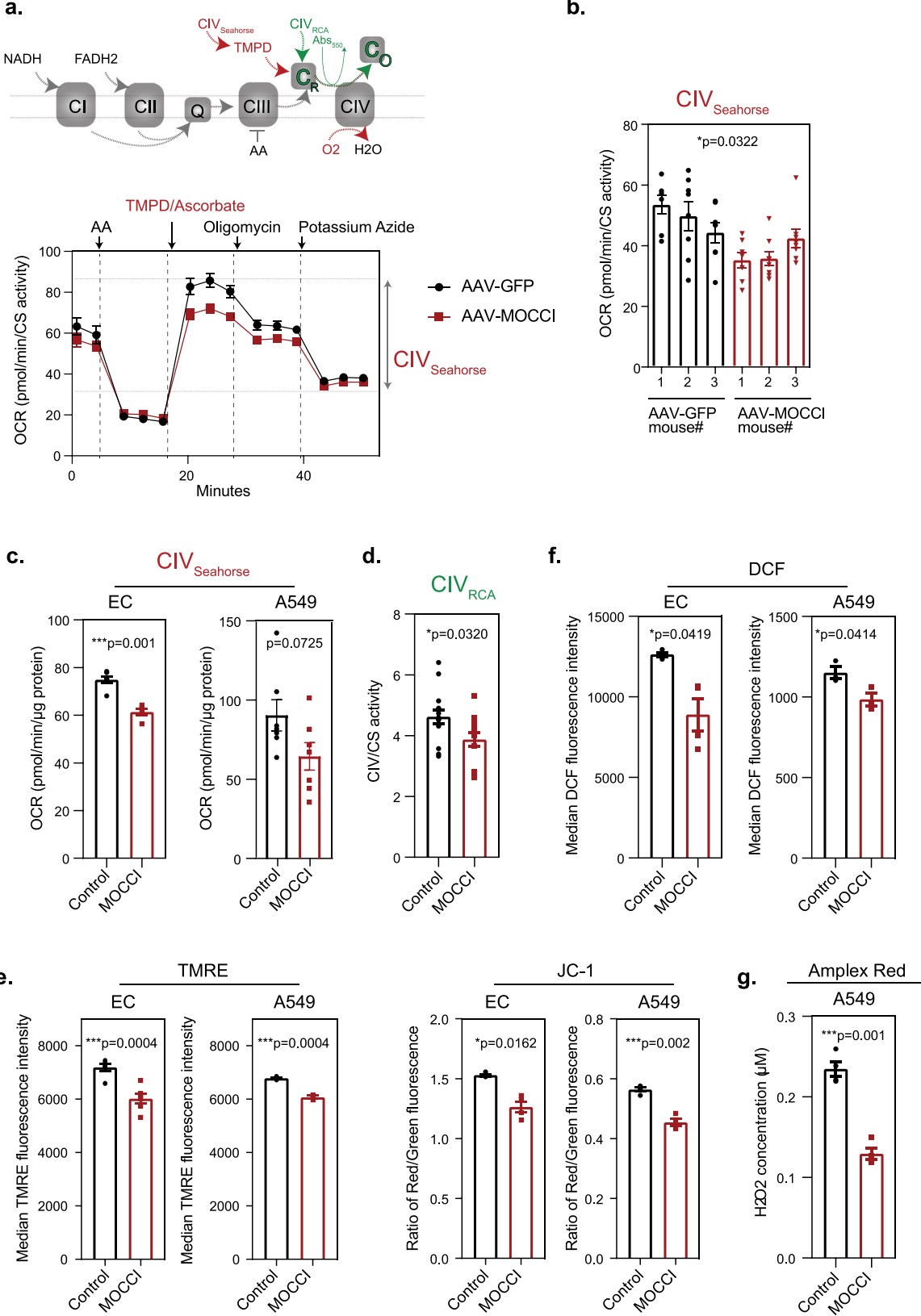

non-coding roles to regulate mitochondrial membrane potential. Of note, knockdown of NDUFA4 did not decrease mitochondrial membrane potential either, suggesting that the incorporation of MOCCI specifically reduces membrane potential and ROS production rather than the removal of NDUFA4 (Supplementary Fig. 6d). These results illustrate a complex scenario with the

following implications. First, the reduction in membrane potential and ROS production in MOCCI HAECs and A549 cells is not simply due to impaired mitochondrial integrity owing to protein overexpression, as WT-mRNA drives MOCCI expression to similar levels compared to MOCCI ORF (Fig. 3f). Secondly, the reductions in membrane potential and ROS generation in

**Fig. 5 MOCCI reduces CIV activity, membrane potential, and ROS production. a** (Top) Schematic uncoupled electron flow assay on Agilent's Seahorse platform to measure CIV activity in isolated mitochondria using TMPD/ascorbate as electron donor (see Methods). (Bottom) Seahorse uncoupled electron flow analysis of AAV-MOCCI and AAV-GFP isolated mouse heart mitochondria. Oxygen consumption rate (OCR) was normalized with citric synthase (CS) activity. Dotted line shows the compounds injected at each time point. CIV activity = maximal activity after TMPD addition − minimal activity after azide injection. AA antimycin A, TMPD N,N,N',N'-Tetramethyl-p-phenylenediamine dihydrochloride. Data were presented as mean and SEM of 3 biological replicates. **b** CIV activity of AAV-MOCCI and AAV-GFP isolated mouse heart mitochondria as measured by uncoupled electron flow assay on Seahorse shown in (**a**). Each column represents one mouse, and each dot represents one technical replicate. Data were presented as mean ± SEM; P = two-tailed unpaired Student's t test; n = 3 biological replicates with 6 technical replicates each. **c** CIV activity of digitonin (0.0025%) permeabilized MOCCI and control cells as measured by Seahorse. Data were presented as mean ± SEM; P = two-tailed unpaired Student's t test; n = 6 technical replicates each. **d** CIV activity of MOCCI and control A549 cells as measured by respiratory chain enzyme assay (RCA). Data were presented as mean ± SEM; P = two-tailed unpaired Student's t test; n = 8 biological replicates each. **e** Membrane potential of MOCCI and control cells as measured by flow cytometry of TMRE or JC-1 dye incorporation. Data are presented as mean ± SEM; P = one-way ANOVA; n = 4 biological replicates per condition from two repeats. **f** Total cellular ROS levels of MOCCI and control cells as measured by flow cytometry of DCF dye reaction. Data were presented as mean ± SEM; P = one-way ANOVA; n = 3 biological replicates per condition. **g** Total cellular ROS levels of MOCCI and control cells as measured by Amplex Red assay. Data were presented as mean ± SEM; P = one-way ANOVA; n = 4 biological replicates per condition.

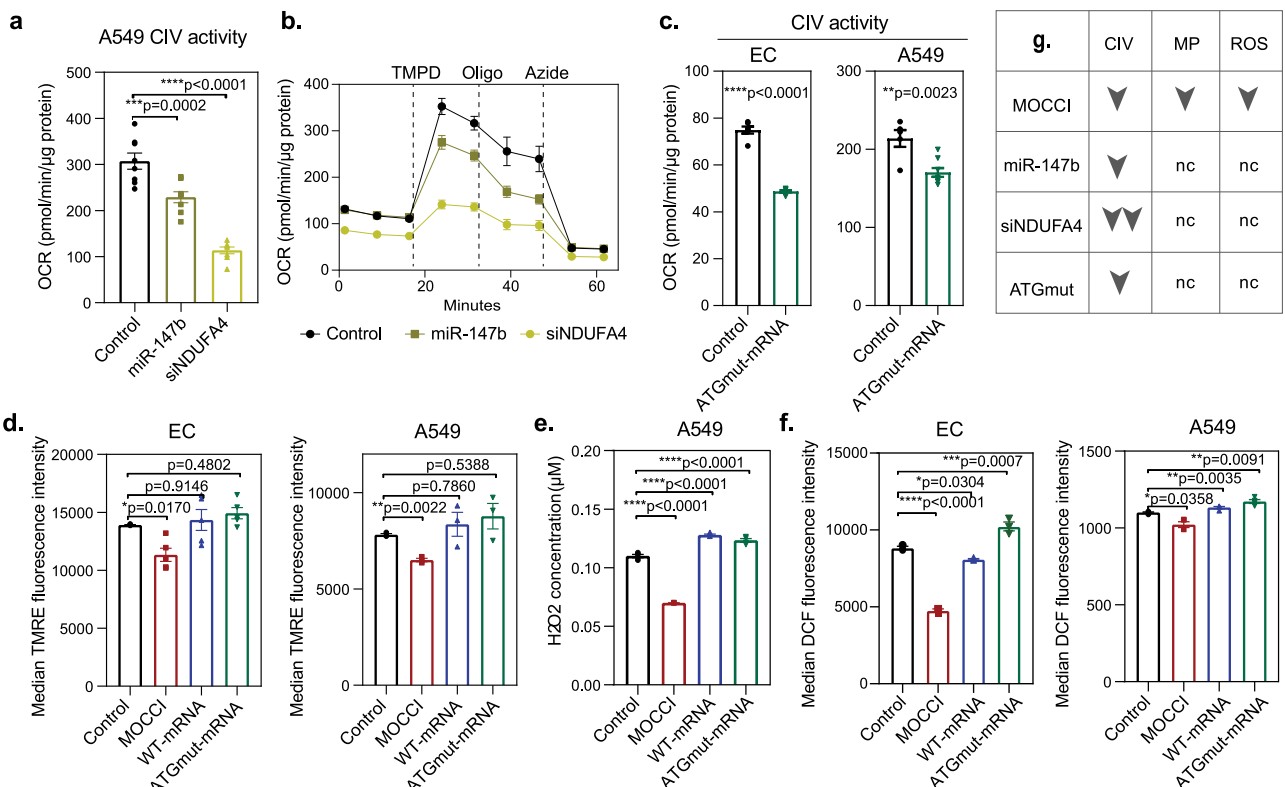

**Fig. 6 miR-147b reduces CIV activity but not membrane potential and ROS production. a** CIV activity of permeabilized A549 cells transfected with miR-control, miR-147b mimic, or siNDUFA4 as measured by Seahorse. Data were presented as mean ± SEM; P = one-way ANOVA; n = 8 technical replicates per condition. **b** Seahorse uncoupled electron flow analysis of CIV activity in permeabilized A549 cells transfected with miR-control, miR-147b mimic, or siNDUFA4, as shown in (**a**). Oxygen consumption rate (OCR) reads were normalized with protein levels quantified with BCA. Dotted line shows the compounds injected at each time point. CIV activity was measured as activity between the maximal activity after TMPD addition and the minimal activity after azide injection. AA antimycin A, TMPD N,N,N',N'-Tetramethyl-p-phenylenediamine dihydrochloride, Oligo oligomycin. Data were presented as mean and SEM of eight biological replicates. **c** CIV activity of permeabilized WT or ATGmut-mRNA overexpressing- HAECs and A549 cells, as measured by Seahorse. Data were presented as mean ± SEM; P = two-tailed unpaired Student's t test; n = 8 technical replicates per condition. **d** Membrane potential of HAECs and A549 cells with indicated transgenes as measured by flow cytometry of TMRE dye incorporation. Data were presented as mean ± SEM; P = one-way ANOVA; n = 4 biological replicates per condition from two repeats. **e** Total cellular ROS levels of A549 cells with indicated transgenes as measured by Amplex Red assay. Data were presented as mean ± SEM; P = one-way ANOVA; n = 4 biological replicates per condition. Data for control and MOCCI were shown in Fig. 5g. **f** Total cellular ROS levels of HAECs and A549 cells with indicated transgene as measured by flow cytometry of DCF dye reaction. Data were presented as mean ± SEM; P = one-way ANOVA; n = 3 biological replicates per condition. **g** Effects of MOCCI, miR-147b mimic, siNDUFA4, and ATGmut-mRNA on CIV activity, membrane potential (MP), and ROS production.

MOCCI cells may not be directly caused by reduction of CIV activity, since miR-147b reduce CIV activity without reducing membrane potential and ROS. Lastly, these results indicate that MOCCI incorporation is not functionally equivalent to removal of NDUFA4 via miR-147b or siNDUFA4, as summarized in Fig. 6g. This provides a conceptual explanation for the presence of miR-147b in the 3′UTR of *C15ORF48*, even though MOCCI alone is sufficient to mediate the NDUFA4-to-MOCCI switch at the protein level. While downregulation of *NDUFA4* mRNA by miR-147b might serve to reinforce the switch, it is likely to have additional roles that MOCCI cannot fulfill. We next sought to define the functional effects of MOCCI and miR-147b expression on the inflammatory response.

**MOCCI and miR-147b reduce pro-inflammatory cytokine production**. To address this question, we first examined the effects of a NDUFA4-to-MOCCI switch mediated by MOCCI alone. *C15ORF48* expression in human cells is upregulated in acute bacterial and viral infections (Supplementary Fig. 7a and Table 1), suggesting a role during an immune response. Even though ECs are not leukocytes, they constitute the first line of defense against bloodborne pathogens, and are prone to excessive inflammation triggered by infections. By clustering genes that have the same expression pattern into modules (see Methods), our IL-1β stimulated HAEC RNA-seq data show that *C15ORF48* is co-expressed with genes in two modules (Modules 2 and 3 in Fig. 7a). These modules are enriched for negative regulation of cytokine production and activated T cell proliferation, and IL-10 production, which represent anti-inflammatory strategies aimed at host protection via limiting excessive immune activation during an infection. We, therefore examined the effects of MOCCI on cytokine production vis-à-vis WT and ATGmut-mRNA in HAECs following a viral challenge. We used the flaviviruses Dengue (DENV) and Zika (ZIKV) due to their ability to infect and replicate in ECs, and induce *C15ORF48* expression (Fig. 7b). HAECs primarily secrete IL-6, IL-8, MCP-1 (CCL2), CXCL10, and Arginase when challenged with DENV/ZIKV. Levels of interferons and other common pro-inflammatory cytokines were undetectable (Supplementary Fig. 7b). MOCCI and WT/ATGmut-mRNA significantly suppressed IL-6 and MCP-1 secretion (Fig. 7c, d), while MOCCI-KO HAECs showed the converse phenotype (Fig. 7e). Transfection of miR-147b mimic and siNDUFA4 also reduced MCP-1 secretion, confirming that the effects observed in WT and ATGmut-mRNA can be attributed to miR-147b (Supplementary Fig. 7c). These results together indicate that MOCCI and miR-147b exert a suppressive effect on pro-inflammatory cytokine production in HAECs. This effect was not limited to viral-induced cytokine production. Rather, we found that MOCCI, WT, and ATGmut-mRNA were also sufficient to lower the secretion of IL-6, MCP-1, and even IL-8 upon challenge with bacterial endotoxin LPS, which was more potent in inducing cytokine secretion in HAECs compared to DENV/ZIKV infections (Fig. 7f). Hence, we conclude that in cultured HAECs and A549 cells, both MOCCI and miR-147b are able to reduce MCP-1 and IL-6 production at basal and when challenged with inflammatory stimuli.

Since both MOCCI and miR-147b lower CIV activity, our findings suggest that CIV attenuation is sufficient to suppress the secretion of MCP-1 and IL-6. This is consistent with recent findings that inhibition of CIV in human PBMCs with potassium cyanide results in widespread suppression of pro-inflammatory cytokine production[24]. We confirmed these findings in HAECs using potassium azide as a CIV inhibitor, which significantly reduced MCP-1 and IL-6 secretion without any cytotoxic effects during a 24 h incubation period (Supplementary Fig. 7d). Hence,

CIV inhibition or attenuation suppresses pro-inflammatory cytokine production. In the case of MOCCI HAECs, reduction of ROS production may also contribute to attenuation of MCP-1 and IL-6 production. Treatment of HAECs with anti-oxidant *N*-acetylcysteine (NAC), which effectively lowers ROS in HAECs (Supplementary Fig. 7e), significantly reduced IL-6 and MCP-1 production at basal and following LPS exposure to levels comparable with that of MOCCI HAECs (Supplementary Fig. 7f). However, since miR-147b, WT, and ATGmut-mRNA suppress cytokine production without reducing membrane potential and ROS, this is unlikely to be the sole mechanism with which CIV attenuation reduces MCP-1 and IL-6 production.

**MOCCI and miR-147b regulate the IR via RIG-I/MDA5**. The co-expression of *C15ORF48* with genes associated with type I interferon (Fig. 7a) prompted us to examine the effects of *C15ORF48* expression on antiviral immunity. Intriguingly, MOCCI has a viral homolog in poxviruses[25], suggesting that it has been evolutionarily selected and co-opted by viruses for immune evasion via modulation of the IR. The canonical pathway of type I IR is the IFN-stimulated gene factor 3 (ISGF3) pathway. ISGF3 is induced by the IFN-regulatory factor IRF9, leading to STAT1/STAT2-mediated activation of genes containing IFN-stimulated response elements (ISREs), also known as interferon-stimulated genes (ISGs)[26]. In line with the observed anti-inflammatory effects of MOCCI, RNA-seq analysis of HAECs 48 h postinfection showed that MOCCI HAECs suppressed the upregulation of DENV and ZIKV infection-induced genes, which are highly enriched for ISGs (Fig. 8a and Supplementary Fig. 8a). Enrichment analyses of differentially regulated ISGs suggest that STAT1/STAT2/IRF1/IRF8/ NF-κB target genes are tonically repressed by MOCCI, as well as genes that are regulated by the newly discovered anti-inflammatory transcription factor ETV7 and microRNA miR-146a (Supplementary Fig. 8a)[27,28]. These data suggest that MOCCI suppresses the IR following viral infection. On the contrary, WT-mRNA and ATGmut-mRNA showed the opposite trend of enhancing IR (Fig. 8a). As the IR is responsible for establishing an antiviral state, we tested if these observed trends resulted in functional consequences for viral replication, as measured by fold change of viral genome (vg) copy number 48 h postinfection. Consistent with the observed trends in IR and ISGs, MOCCI HAECs had a higher vg copy number while *C15ORF48* WT-mRNA HAECs and ATGmut-mRNA had lower vg copy number 48 h postinfection, with ATGmut-mRNA-mediated suppression reaching statistical significance (Fig. 8b). Importantly, we did not detect significant differences in vg copy number 2 h postinfection (Supplementary Fig. 8b), arguing that viral entry was not significantly affected. A similar trend was observed in A549, where MOCCI enhanced and ATGmut-mRNA suppressed viral copy number, with WT-mRNA showing an intermediate phenotype (Fig. 8c). Viral replication results correlated well with the levels of *IFN-β* mRNA in A549 cells, a measure of the magnitude of IR, which were repressed by MOCCI and enhanced by WT-mRNA (Supplementary Fig. 8c). Other ISGs were significantly enhanced by ATGmut-mRNA (Supplementary Fig. 8d). These data suggest that miR-147b exerts a viral-static effect that keeps viral replication in check by stimulating the IR. Indeed, transfection of miR-147b was sufficient to inhibit ZIKV replication (Fig. 8d and Supplementary Fig. 8e). Importantly, the antiviral effects of miR-147b are recapitulated by the transfection of siNDUFA4 (Fig. 8d) at levels that match miR-147b-mediated *NDUFA4* mRNA downregulation (Supplementary Fig. 8f). These results indicate that miR-147b enhance the IR by downregulating *NDUFA4* in the absence of replacement by MOCCI. MOCCI incorporation on the other hand reduced cell death following

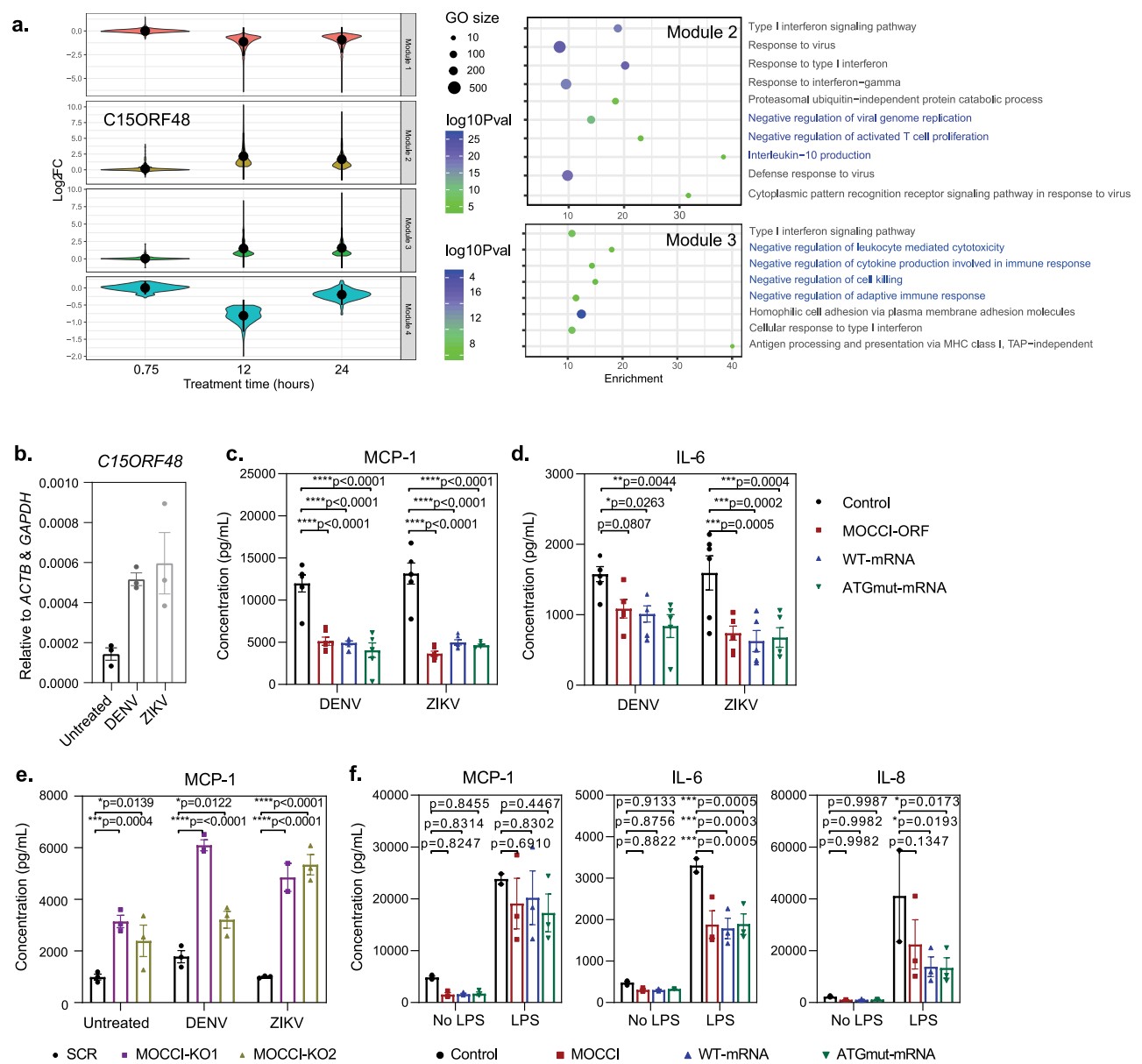

**Fig. 7 MOCCI and miR-147b reduce MCP-1 and IL-6 secretion during viral infection. a** (Left) Differentially expressed genes in the HAECs treated with IL-1β across the timepoint series (Fig. 1b) were clustered into co-expression modules by dynamic tree cut. The log-fold change is relative to the untreated cells. *C15ORF48* resides in Module 2. (Right) Panther GO analysis of genes in Module 2 and closely related Module 3 reveal significant enrichments in the ontologies depicted. Size of dot indicates the number of members in each GO annotation and the color indicates the *p* value of enrichment. **b** *C15ORF48* mRNA levels in HAECs following DENV and ZIKV infection. Data were presented as mean ± SEM; *n* = 3 biological replicates per condition. **c** Levels of MCP-1 (CCL2) secreted by HAECs with indicated transgenes following DENV/ZIKV infection. Data were presented as mean ± SEM; *P* = two-way ANOVA; *n* = 6 biological replicates per condition. **d** Levels of IL-6 secreted by HAECs with indicated transgenes following DENV/ZIKV infection. Data were presented as mean ± SEM; *P* = two-way ANOVA; *n* = 6 biological replicates per condition. **e** Levels of MCP-1 (CCL2) secreted by WT and MOCCI-KO HAECs at basal and following DENV/ZIKV infection. Data were presented as mean ± SEM; *P* = two-way ANOVA; *n* = 3 biological replicates per condition. **f** Levels of MCP-1 (CCL2), IL-6, and IL-8 secreted by HAECs with indicated transgenes at basal and following LPS treatment after 24 h. Data were presented as mean ± SEM; *P* = two-way ANOVA; *n* = 3 biological replicates per condition.

viral infection in both HAECs and A549 (Fig. 8e). Besides suppressing pro-inflammatory cytokine production, MOCCI therefore exerts a cytoprotective effect, albeit at the cost of a small elevation in viral replication.

Lastly, we sought to address the mechanism of IR regulation by MOCCI and miR-147b. Because NDUFA4-to-MOCCI switch lowers ROS production, we examined if MOCCI suppresses the IR through ROS reduction. Contrary to our expectation, NAC treatment of HAECs and A549 reduced vg copy number (Supplementary Fig. 9a). MOCCI-miR-147b regulation of host

antiviral response is unlikely to be directly related to ROS modulation. Hence, we sought to find additional mechanisms of miR-147b antiviral immunity. Following DENV and ZIKV infection, viral RNA genomes are detected by cytosolic RNA sensors RIG-I and MDA5[29], leading to the activation of the NF-κB and IRF3/7, transcriptional mediators of the IR (Fig. 9a)[30]. Following ZIKV infection, both ATGmut HAECs and A549 expressed higher levels of RIG-I and its downstream adapter MDA5 (Fig. 9b). WT-mRNA had higher levels than control but lower levels than ATGmut (Fig. 9b), correlating with the

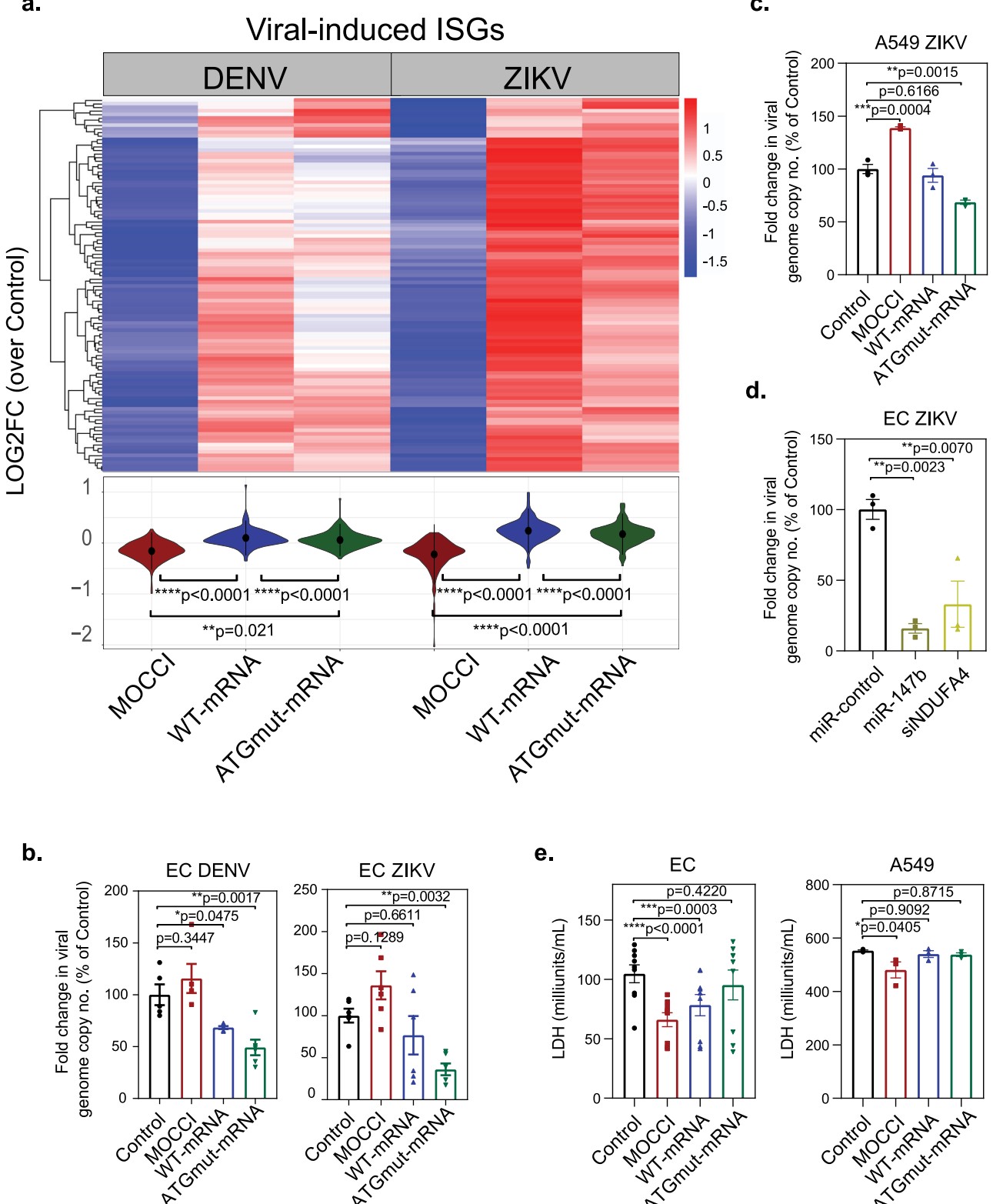

intermediate suppression of viral replication (Fig. 8b, c) and once again highlighting the antagonism between MOCCI and miR-147b with respect to the IR. Downstream of RIG-I and MDA5, ATGmut-mRNA cells had increased NF-κB activation as measured by phosphorylated p65 (Fig. 9b). Despite the enhancement in RIG-I levels, WT and ATGmut-mRNA cells maintain their ability to suppress pro-inflammatory MCP-1 production in

response to a RIG-I agonist (Supplementary Fig. 9b). Altogether, miR-147b potentiates RIG-I/MDA5/NF-κB activity, enhancing the IR and providing host protection during a viral infection. MOCCI on the other hand dampens this response and prevents excessive cell death and hyperinflammation. Altogether, during an acute infectious challenge, *C15ORF48* relies on both its coding and non-coding functions to finetune the balance between

**Fig. 8 MOCCI and miR-147b coordinate to optimize host protection. a** Heatmap of a subset of DENV and ZIKV-induced genes that were downregulated in MOCCI-expressing cells compared to control, based on RNA-seq (see "Module Detection" in Methods for more details). $P$ = one-way ANOVA; $n$ = 3 biological replicates per condition. MOCCI = MOCCI ORF only; WT-mRNA = MOCCI + miR-147b; ATGmut-mRNA = miR-147b only. **b** HAECs with the indicated transgenes were infected with DENV and ZIKV (MOI = 1.0). Forty-eight hours later, viral replication was measured by qPCR-mediated viral genome quantification, compared to a baseline obtained at 2 h postinfection. Values are expressed as percentage of the control of each biological replicate. Data were presented as mean ± SEM; $P$ = one-way ANOVA; $n$ = 6 biological replicates per condition. **c** A549 cells with the indicated transgenes were infected with DENV and ZIKV (MOI = 0.1). Fold change was calculated as the same in "**c**". Data were presented as mean ± SEM; $P$ = one-way ANOVA; $n$ = 3 biological replicates per condition. **d** HAECs transfected with *miR-control*, *miR-147b* mimic, or *siNDUFA4* were infected with ZIKV (MOI = 1.0). Fold change was calculated as the same in "**c**". Data were presented as mean ± SEM; $P$ = one-way ANOVA; $n$ = 3 biological replicates per condition. **e** Quantitation of cell death in HAECs and A549 cells with indicated transgenes following ZIKV infection as measured by lactate dehydrogenase (LDH) release. Data were presented as mean ± SEM; $P$ = one-way ANOVA; $n$ = 6 biological replicates per condition from two repeats for HAECs and 3 biological replicates per condition for A549 cells.

inflammation and host protection by combining the anti-inflammatory and cytoprotective effects of MOCCI and the antiviral effects of miR-147b (Fig. 9c). These two activities potentially lead to synergistic protection of the host during viral infection, on the one hand controlling viral replication and on the other controlling host production of pro-inflammatory cytokines and cell death that cause tissue damage.

## Discussion
By intersecting with cytosolic sensors of danger and damage such as RLRs and NLRs, mitochondrial integrity and functionality are key determinants of inflammatory activation[31]. Rising from our observations that mitochondrial SEPs display a programmatic anti-association with inflammatory gene signatures (Fig. 1a), we performed an unbiased proteogenomic screen to uncover mito-SEPs that regulate inflammation. This led to the identification of *C15ORF48* as the top inflammatory mito-SEP or "*i*-Mito-SEP". The *C15ORF48* gene is also known as *NMES-1* as a cancer-associated gene with uncharacterized function[32]. Altogether, results uncover a mitochondrial program during acute inflammation induced by the mito-SEP which we name MOCCI encoded by the *C15ORF48* gene. Activation of this gene following viral infection or pro-inflammatory stimulus IL-1β induces switching of the 14th subunit of CIV from NDUFA4-to-MOCCI, accompanied by the biogenesis of miR-147b in the cytosol (Fig. 9c). MOCCI tonically dampens CIV activity leading to lowered membrane potential and reduced ROS production. Lowered ROS levels, in turn, contribute to suppressing pro-inflammatory cytokine production and the IR, but this is mitigated by the strong antiviral activity of miR-147b. In this fashion, the synergistic and complementary roles of MOCCI and miR-147b in HAECs together provide maximal host protection following an acute viral infection by suppressing host inflammation and viral replication.

The respiratory chain consists of a series of five multi-subunit ETC complexes encoded by 13 genes located in mitochondrial (mt) DNA and over 80 genes in the nuclear DNA. The ETC resides in the IMM and is responsible for over 80% of cellular ATP production. Traditionally viewed as invariant machines that perform housekeeping functions, it is increasingly apparent that ETC complexes can have variable compositions across tissue types and cellular contexts. For instance, CI is comprised of 45 subunits, of which more than half are accessory subunits performing non-enzymatic roles, potentially in assembly and stability of CI and the respiratory chain supercomplexes[33]. The expression levels of some subunits vary across tissues, suggesting that CI assemblies across cell types are not invariant. In the same manner, CIV which contains 14 subunits and over 20 assembly factors, is unique among ETC complexes in that several of its core subunits have isoforms that are expressed in a developmental stage-specific or oxygen-responsive manner[34]. These CIV isozymes are thought to

differ in their oxidation potential and proton pumping efficiency[35] in response to respiratory control cues. For instance, since oxygen is a substrate of CIV, the activity of the CIV must be directly coupled to and modulated by the availability of oxygen. This is achieved through oxygen-regulated expression of CIV subunit variants, namely COX4-1 and COX4-2. While COX4-1 expression is ubiquitous, COX4-2 is highly specific to the lung epithelium and induced upon hypoxic stress, where it increases the activity of CIV and optimizes its efficiency including reducing ROS under low oxygen availability[36]. Similarly, COX7A2 is exchanged for COX7A1 under hypoxic stress[37]. Assembly factors of CIV, such as the hypoxia-induced HIGD1A, have also been reported to play a role in boosting CIV activity when oxygen is limiting[38]. The literature is hence requisite with examples of CIV isozymes being employed to meet cellular energetic requirements under distinct cellular contexts with respect to oxygen availability. Because of the prominent interactions between the ETC and OXPHOS with other areas of cellular biology such as the control of cell death and inflammation, an outstanding question is whether CIV isozymes are also involved in the control of such functions. MOCCI was previously identified by Pagliarini and colleagues in a high throughput interactome screen as a mitochondrial uncharacterized (x) protein (MXP) that interacts with components of the ETC[14]. Here, we demonstrate that MOCCI is a close paralog of NDUFA4. NDUFA4 was previously thought to be a subunit of CI, and later reclassified to be the 14th subunit of mature CIV complex[15–17]. Its controversial assignment as an integral CIV subunit, owing to its absence in several solved structures of bacterial, fungal, and mammalian CIV, can be explained by its sensitivity to cholate detergents[17], which remove NDUFA4 from the rest of the complex. NDUFA4 is not required for CIV assembly[15,16], but rather has been speculated to enhance CIV activity by preventing the dimerization of CIV monomers[17]. While NDUFA4 expression is constitutive and ubiquitous, MOCCI is induced transcriptionally and translationally upon inflammatory stimulation by IL-1β. We demonstrate that MOCCI is incorporated into mature CIV monomer and dimers, and to a smaller extent, supercomplexes. Furthermore, its inclusion into CIV occurs concurrently with the downregulation of NDUFA4, with which it shares an almost superimposable predicted secondary structure. On this basis, supported by the finding that MOCCI and NDUFA4 are not found in the same CIV complexes, we propose that NDUFA4 is replaced by MOCCI in what we term the NDUFA4-to-MOCCI switch. MOCCI is therefore a variant of the 14th subunit of cytochrome C oxidase. Whether NDUFA4 is actively displaced by MOCCI in existing CIV complexes, or CIV-NDUFA4 is replaced by newly synthesized CIV-MOCCI is currently unclear. While this paper was in preparation, Kurihara and colleagues reported their findings that MOCCI, which they named COXA4L3, is expressed in developing spermatids where it is

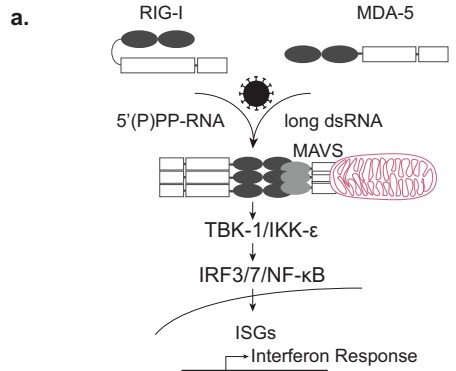

**a.**

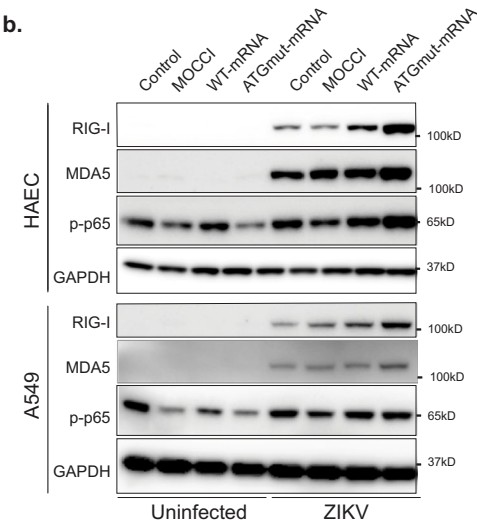

**b.**

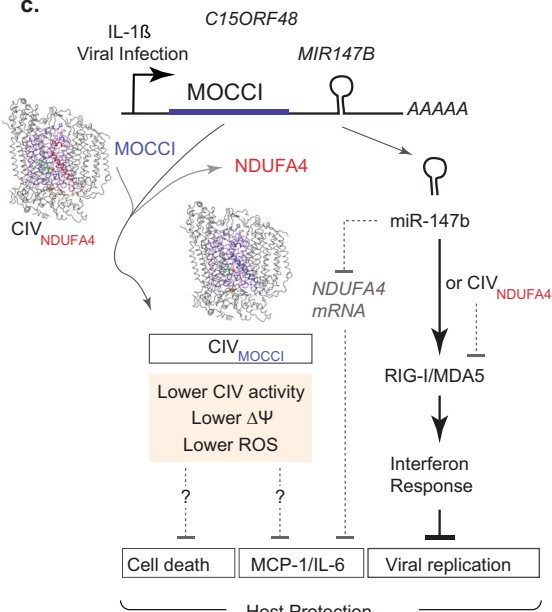

**c.**

**Fig. 9 miR-147b works through RIG-I/MDA5 pathway to modulate interferon response. a** RIG-I/MDA5 pathway of viral RNA recognition and interferon response. ISG interferon-stimulated genes. **b** Western blot of RIG-I, MDA5, and phosphorylated p65 in uninfected and 24 h ZIKV-infected HAECs and A549 cells with the indicated overexpression. $n = 2$ biological replicates. Source data are provided as a Source Data file. **c** Model of how coding and non-coding functions of *C15ORF48* transcript coordinate to provide host protection during viral infection. The exact mechanism on how inhibiting CIV activity can modulate cytokine secretion and cell death remains an outstanding question.

attenuation of CIV activity by both MOCCI and miR-147b results in lowered cytokine production. How CIV activity controls cytokine production is an open area of investigation. One possible mechanism, as seen with MOCCI expression, is the reduction of ROS production, but that does not appear to be the case since miR-147b suppresses cytokine production without reducing ROS. CIV inhibition could modulate cytokine production indirectly through retrograde signaling pathways that are activated during mitochondrial stress[40]. A growing consensus points to mitochondrial retrograde signaling as a type of integrated stress response[41,42]. An intriguing possibility is that changes in CIV activity are sensed and relayed to downstream effectors through similar mechanisms.

How the NDUFA4-to-MOCCI switch leads to dampening of CIV activity is also unclear, as neither MOCCI incorporation nor NDUFA4 depletion affects total CIV levels. One possibility is based on observations that CIV transitions between monomeric and dimeric forms[43,44], which assemble distinct COX7 paralogues COX7A2 and COX7A, respectively[45]. CIV dimers have lower activity compared to monomers[46] and are sensitive to inhibition by high ATP/ADP ratios. Dimerization of CIV monomers has been documented to lower CIV activity and is postulated to reduce ROS production[44]. Based on the structural observation that NDUFA4 resides in the interface between CIV monomers, Yang and colleagues postulated that NDUFA4 inhibits dimerization of CIV, thereby maintaining CIV in a maximally active form[17]. MOCCI incorporation and NDUFA4 depletion might favor CIV dimerization and in so doing lower CIV activity and ROS production. However, our BN-PAGE analyses do not provide evidence to support this hypothesis, potentially due to the non-physiological nature of detergent extraction and the need for more in-situ physiological methods such as structural comparisons of MOCCI versus NDUFA4-containing CIV complexes.

It is important to point out that the reductions in CIV caused by MOCCI versus miR-147b are not qualitatively the same. In the case of MOCCI, reduction of CIV activity is accompanied by lower membrane potential and reduced ROS generation. This is not the case for miR-147b. These results make it clear that NDUFA4 removal with or without replacement by MOCCI have different effects downstream of dampening CIV activity, for instance in differential regulation of the IR. How MOCCI lowers membrane potential and ROS production are open questions. One possibility is that MOCCI incorporation alters the intrinsic uncoupling of CIV, also known as "proton slip", which is the regulated diminishing of the proton pumping efficiency of the ETC complexes I, III, IV, and V such that the $H^+/e^-$ ratio falls below $1.0$[47–49]. Among all ETC complexes, regulated proton slip is only considered to be physiologically relevant at CIV[50] and is thought to confer respiratory control at high $\Delta\Psi_m$ to prevent excessive ROS generation[51]. It has been observed that tissue-specific isoforms of CIV containing COX6A-1/2 have different $H^+/e^-$ ratios, suggesting that other non-enzymatic subunits that exhibit specific expression patterns (such as MOCCI) might

incorporated into CIV[39], providing independent confirmation that MOCCI is a CIV protein.

The switch from NDUFA4-to-MOCCI downregulates the activity of CIV. miR-147b-mediated NDUFA4 downregulation alone also reduces CIV activity, in line with previous findings[15]. On the basis that CIV inhibition can reduce pro-inflammatory cytokine production in human PBMCs[24], we postulate that

contribute to regulating the $H^+/e^-$ ratio of CIV in distinct cellular contexts[52] to modulate membrane potential and ROS generation. Of note, NDUFA4L2, the other paralog of NDUFA4, also exhibits inducible expression by hypoxia, and is thought to inhibit the activity of Complex I, reducing flux through the ETC and thereby reducing ROS generation[53]. Hypoxia activates pro-inflammatory gene expression[54]. The induction of NDUFA4L2 upon hypoxia and the ensuing decrease in ROS production might, in a similar way as what we observe with MOCCI, serve to dampen hypoxia-induced inflammation. MOCCI may also function in a similar manner during IL-1β challenge to induce immune-dampening responses through uncoupling.

The 3′UTR of *C15ORF48* contains the conserved microRNA *miR-147b*, indicating a strong evolutionary selection to keep both genes on the same transcript (Fig. 2a). Mature miR-147b-3p has previously been characterized as an anti-inflammatory microRNA in monocytes[55], in agreement with our findings that MOCCI exerts an anti-inflammatory effect. In addition, the top predicted target of miR-147b is *NDUFA4*, which we validated experimentally here. A simple scenario is one which MOCCI and miR-147b function as a pair by mutually reinforcing the NDUFA4-to-MOCCI switch. Our data however indicate that MOCCI and miR-147b have both convergent and divergent functions. On one hand, both MOCCI and miR-147b lower CIV activity and reduce MCP-1 and IL-6 production. On the other hand, MOCCI reduced membrane potential and reduced ROS production, while miR-147b alone potentiated the IR pathway by increasing RIG-I levels. In both of these processes, MOCCI and miR-147b are mutually antagonistic i.e., miR-147b opposes MOCCI-mediated drop in ROS and membrane potential (Fig. 6d–f), while MOCCI opposes miR-147b enhancement of RIG-I (Fig. 9b). Consequently, the phenotypic effects of expressing MOCCI are not the same as that of expressing ATGmut-mRNA, and expression of the WT-mRNA had intermediate effects. For instance, while MOCCI suppressed the IR response, leading to increased viral replication following DENV/ZIKV infection, WT-mRNA and ATGmut-mRNA enhanced the IR leading to suppression of viral replication. Altogether, these data indicate that MOCCI and miR-147b, while potentially working in concert to effect a NDUFA4-to-MOCCI switch in CIV, are not functionally equivalent. Whilst the IFN-dampening effect of MOCCI may serve to limit immune pathology, it also potentially favors viral replication. This is in turn counteracted by miR-147b, which functions to control viral replication. In the presence of both MOCCI and miR-147b, WT-mRNA HAECs dually control viral replication and mitigate pro-inflammatory cytokine production.

Notably, our model assumes that both MOCCI and miR-147b are present in the same cell that expresses *C15ORF48* mRNA. According to existing paradigms, processing of exonic miRNAs like miR-147b causes mRNA destabilization due to poly-A removal—a requisite step in miR biogenesis[56]. mRNA translation and processing of pre-pri-mRNA should in theory be mutually exclusive. Therefore, it is possible that endogenous MOCCI and miR-147b are not usually present in the same cell at the same time. Rather, they are active in different subsets of cells within a tissue where they play distinct functions. For instance, in human PBMCs, *C15ORF48* mRNA is expressed constitutively at high levels in monocytes (Supplementary Fig. 2a). Results from our THP-1 model in vitro predict that these monocytes express high levels of miR-147b (Supplementary Fig. 5a–d), consistent with their main function of immune surveillance and viral defense. Endothelial cells, on the other hand, induce *C15ORF48* and translate the MOCCI peptide efficiently due to the exquisite need for protection against ROS production, which leads to endothelial dysfunction and further vascular damage. An outstanding question is the mechanism that govern cell-type specific translation of

MOCCI and biogenesis of miR-147b. Altogether, our study highlights how coding and non-coding functions of a gene can be elegantly coordinated to achieve a specific biological outcome.

In conclusion, starting from an initial observation that mito-SEPs correlate negatively with inflammation, we present here our finding that *C15ORF48* and the mito-SEP it encodes—MOCCI—assembles a CIV isozyme that fine-tunes the ETC to dampen inflammation following acute infection. Our results suggest that the MOCCI-miR-147b axis or modulating CIV activity in general might be a feasible strategy for ameliorating viral-induced hyperinflammation, a possibility that calls for further investigation. Additionally, since our study only addressed the effects of *C15ORF48* in acute inflammation, it will be interesting to investigate its ability to ameliorate chronic inflammation particularly in the setting of cardiovascular diseases. Lastly, this study should motivate further efforts to mine the mito-SEP peptidome for other regulators of the respiratory chain that function to coordinate a ubiquitous biological process of oxidative phosphorylation to achieve specific, context-dependent cellular outcomes.

## Methods

**Cell culture**. Primary HAECs (PromoCell® C12271) were cultured in endothelial cell growth media MV2 (PromoCell® C22022) with changes every 2 days. To subculture, HAECS were washed once with PBS, trypsinized (0.25% trypsin; Life Technologies 25200072) for 90 s, neutralized with an equal amount of EGMV2 media, and centrifuged at 500xg for 5 min, resuspended and seeded at a density of 5000 cells/cm². For regular maintenance, plates were coated with 7 µg/mL fibronectin (Promocell C-43050) for an hour before they were rinsed with PBS. This coating is omitted during experimental set ups to prevent any confounding effects.

A549, HEK293T, HeLa, and U87MG cells were cultured in 10% FCS/DMEM (Hyclone SH30022.01). NTERT and pulmonary artery smooth muscles cells were cultured in Keratinocyte serum-free media (Life Technologies 17005042) and SmGMTM-2 smooth muscle cell growth medium-2 (Lonza Bioscience CC-3182), respectively. They were subcultured with trypsin in the same way as HAECs.

THP-1 cells were cultured in 10% FCS/RPMI (Hyclone SH30255.01). They were maintained at a density between 100,000–1,000,000 cells/mL. To differentiate them to M1 macrophages, 800,000 cells/well were seeded into 12-well plates in 50 nM PMA (Sigma-Aldrich P1585). After 48 h, the media was replaced with 10% FCS/RPMI for 24 h. The cells were then differentiated into M1 macrophages in media containing 100 ng/mL LPS (InvivoGen tlrl-3pelps) and 20 ng/mL IFNg (Miltenyi Biotec 130-096-481). After 8 h, 1 ng/mL IL-1β or water were added into the appropriate wells while still in the presence of LPS and IFNg for 16 h before the cells were harvested for flow cytometry or RNA.

The concentrations of cytokines used are the following, unless otherwise stated: 1 µg/mL LPS (InvivoGen tlrl-3pelps) and 1 ng/mL IL-1β (Sigma-Aldrich SRP3083). Potassium azide (Sigma-Aldrich 740411) was used at the concentrations indicated in the figure.

**Generation of MOCCI overexpressing cells**. Overexpression of MOCCI in HEK293T was achieved by transfecting them with pCS2+ -MOCCI, which drives MOCCI ORF expression using a CMV promoter. The ORF was amplified by PCR from cDNA and inserted into a pCS2+ backbone, under the control of a CMV promoter. Overall 4,000,000 HEK293T cells were seeded per poly-L-lysine (Sigma-Aldrich P4707) coated 10 cm dish. After 16 h, 7 µg of plasmid per plate was transfected using 21 µL of Lipofectamine 2000 (Life Technologies 11668019). The media was changed out after 8 h and the cells were harvested after 48 h.

Overexpression in HAECs and A549 was achieved by lentiviral transduction. Transfer plasmids pCDH-*C15ORF48*-ORF (contains *MOCCI* ORF), pCDH-*C15ORF48*-mRNA (contains WT-mRNA), and pCDH-*C15ORF48*-ATGmut-mRNA (contains ATGmut-mRNA) were used to produce lentivirus. Control cells were transduced with pCDH-control which contains a scrambled ORF in place of the MOCCI constructs. MOCCI ORF and *C15ORF48* mRNA were amplified from cDNA and inserted into a pCDH vector containing Neomycin resistance cassette. ATGmut-mRNA (*C15ORF48* mRNA with any in-frame "ATG" mutated to "ATC") was synthesized and inserted into the same backbone. Transfer plasmids were co-transfected with packaging plasmids into Lenti-X cells to generate lentiviral particles. Virus-containing media were collected 2 days and 3 days after transfection. The lentiviral particles were concentrated using Amicon® Ultra-15 Centrifugal Filter Unit (Sigma-Aldrich UFC910024) and kept frozen at −80 °C. Equal titer of virus was added onto the cells with 8 µg/mL polybrene (Sigma-Aldrich 9268) and changed out the next day. After 3 days, cells containing the vector were selected for in 500 µg/mL G418 (Santa Cruz sc-29065A). As endothelial cells have limited lifespan, plasmids instead of cells are available upon request.

**Generation of MOCCI knockout cells.** MOCCI-knockout (KO) cells were generated using lentiCRISPRv2 transfer plasmid (Addgene plasmid 52961; a gift from Feng Zhang)[57]. The gRNA sequences for lentiCRISPRv2-C15ORF48-gRNA1 and lentiCRISPRv2-C15ORF48-gRNA2 were: 5′-TTATAACAATCAACCAACAA-3′ and 5′-TTCATGACTGTGGCGGCGGG-3′. These gRNAs target the open reading frame of the gene and do not target *MIR-147b*. Endothelial cells were transduced with lentivirus containing the gRNAs for 24 h. After 3 days, cells containing the cassette were selected for with 1 μg/mL puromycin (InvivoGen ant-pr-1). After a week, the cells were stimulated with 1 ng/mL IL-1β and checked for MOCCI expression by intracellular flow cytometry. To verify the gene disruption, primers binding to the 5′UTR and 3′UTR of *C15ORF48* were used to amplify the transcript from cDNA and the amplicons were sent for sanger sequencing. The efficiency of editing was inferred by sequence trace decomposition using TIDE[58]. As endothelial cells have limited lifespan, plasmids instead of cells are available upon request.

**Transient overexpression of *miR-147b* mimic and *siRNA*.** Overall 500,000 endothelial cells or 1,000,000 HEK293T were plated per 6 cm dish. The next day, *miR-147b* mimic, *miR-SCR* control, or *siNDUFA4* (Dharmacon C-301239, CN-001000, L-019200-01) were transfected into the cells using 2 μL of Dharmafect Reagent 1. 25 nM final concentration of RNA was used in the experiment to verify the knockdown of NDUFA4 by miR-147b. Five nanomolar of RNA was used in the subsequent experiments to achieve similar knockdown by ATGmut-mRNA construct. After 8 h, the media was replaced with fresh media. To check for expression of *NDUFA4* mRNA and NDUFA4 protein, the cells were harvested after 2 days. For functional readouts, the cells were harvested 1 day post-transfection and seeded for experiments.

**Adeno-associated virus (AAV) production, purification, and administration.** Viruses were produced as per standard protocol[59]. AAV-mMOCCI and AAV-GFP contain the ORF in *AA467197*, the murine homolog of *C15ORF48* and GFP driven by CMV promoter, respectively. Briefly, AAVs were packaged via triple transfection of HEK293. HEK293 were seeded in 10% fetal bovine serum (FBS)/DMEM supplemented with glutaMax (ThermoFisher Scientific 35050061), pyruvate, (ThermoFisher Scientific 11360070), and MEM non-essential amino acids (Gibco 1140050). Confluency at transfection was between 70–90%. Media was replaced with fresh pre-warmed growth media before transfection. For each HYPERFlask "M" (Corning CLS10034), 300 μg of pHelper (Cell Biolabs), 150 μg of pRepCap (encoding AAV9 (UPenn Vector Core)), and 150 μg of pAAV (containing the ITR-cargo-ITR) were mixed in 7 ml of DMEM, followed by mixing with 2.8 mg of PEI "MAX" 40k (Polysciences 24765-1). The mixture was incubated at room temperature for 15 min, and transferred drop wise to the cell media. The day after transfection, the media was changed to 2% FBS/DMEM/glutaMax/pyruvate/MEM non-essential amino acids. Cells were harvested 48–72 h after transfection by dissociation with 5 mM EDTA/PBS (pH 7.2), and pelleted at 1500 g for 12 min. Cell pellets were resuspended in 5 ml of lysis buffer (Tris-HCl pH 7.5, 2 mM MgCl$_2$, 150 mM NaCl), and freeze-thawed three times between dry-ice-ethanol bath and 37 °C water bath. Cell debris was clarified via 4000x*g* for 5 min, and the supernatant collected. The collected supernatant was treated with 50 U/ml of Benzonase (Sigma-Aldrich E8263) and 1 U/ml of RNase cocktail (Invitrogen 10638255) for 30 min at 37 °C to remove unpackaged nucleic acids. After incubation, the lysate was loaded on top of a discontinuous density gradient consisting of 4, 6, 7, and 4 ml of 15, 25, 40, and 60% Optiprep (Sigma-Aldrich D1556), respectively in a 29.9 mL Optiseal polypropylene tube (Beckman-Colter 361625). The tubes were ultracentrifuged at 213,918x*g*, at 18 °C, for 1.5 h, on a Type 70 Ti rotor. The 40% fraction was extracted, and dialyzed with 0.001% pluronic acid/PBS, using Amicon Ultra-15 (100 kDa MWCO) (Millipore UFC910024). The titer of the purified AAV vector stocks were determined using real-time qPCR with ITR-sequence-specific primers and probe[60], referenced against the ATCC reference standard material 8 (ATCC).

To inject the AAV into the intrathoracic cavity, P6 C57BL/6 J pups were placed in a latex sleeve and cryoanesthised from neck down in crushed ice and water for 5–8 min. A total of $6 \times 10^9$ viral particles were then injected using a 31 G hypodermic needle. After the injection, the pups were placed under a heat lamp to recover before they were returned to their mothers. Heart tissue was extracted for analysis 8–16 weeks following injection.

All animal protocols were approved by the NUS Institutional Animal Care and Use Committee (IACUC). All mice were exposed to a 12-hour light/12-hour dark cycle with controlled humidity and temperature at 23 °C.

**RNA extraction.** RNA was extracted using MN Nucleospin RNA kit (Macherey-Nagel 740955.50) according to the manufacturer's instructions.

**Quantitative PCR.** mRNA was reverse transcribed using iScript RT Supermix (Bio-Rad Laboratories 1708841). Quantitative PCR was performed with iTaq™ Universal SYBR® Green Supermix (Bio-Rad Laboratories 1725124) using gene specific primers. The sequences of the primers are listed in Supplementary Table 1. Primers were purchased from integrated DNA Technologies, Inc. (IDT).

miRNA was reverse transcribed with the TaqMan™ MicroRNA Reverse Transcription Kit (ThermoFisher Scientific 4366596). Quantitative PCR was performed with TaqMan™ Universal Master Mix II, no UNG (ThermoFisher Scientific 4440048) and acquired in a 7900 SDS system. The gene specific primers for reverse transcription and qPCR were from ThermoFisher Scientific (4427975; 4440886).

**Preparation of RNA-seq libraries and downstream analysis.** mRNA was enriched from 1 μg total RNA using NEBNext® Poly(A) mRNA Magnetic Isolation Module (New England Biolabs E7490L). The library was prepared from the mRNA enriched fraction using NEBNext® Ultra™ II Directional RNA Library Prep (New England Biolabs E7765L) according to the manufacturer's instructions. The library was sequenced on the Illumina HiSeq 2500 platform, generating 150 bp end reads. The reads were trimmed for adapters using Trimmomatic v0.36[61] and aligned to GRCh38 genome assembly using STAR aligner v2.6[62] and gene-wise read coverage was quantified using Feature Counts from the Subread package v1.63[63]. All downstream analysis was performed using R 3.5.1[64]. Genes with TPM below 1 in every sample were discarded. DESeq2[65] was used to identify differentially expressed genes. Gene ontology pathway analysis were done using DAVID v6.8[66,67], Enrichr[68,69], and Panther v15.0[70].

**Preparation of Ribo-seq libraries and identification of open reading frames.** About 4,000,000 HAECs from a 10 cm dish were harvested for each sample. An identical set of cells were harvested for RNA-seq. Cell pellets for Ribo-seq were snap-frozen in liquid nitrogen, and then homogenized in 1 mL mammalian lysis buffer (Truseq® Ribo Profile, Illumina RPHMR12126). Lysates were then centrifuged to clear the debris and footprinted with Truseq® Nuclease Select for ribosomal bound RNA. Ribosomal protected fragments were then isolated using Illustra Microspin Sephacryl S400 columns (GE Healthcare) and extracted using phenol:chloroform:isoamylalcohol. Ribosomal RNA was removed using Ribo-Zero Gold rRNA Removal Kit (Illumina) and the library was prepared using Truseq® Ribo Profile (Illumina). The libraries were sequenced on the Illumina HiSeq 2500 platform, generating 50 bp single end reads.

The adapters were trimmed using Trimmomatic v0.36[61] and reads that aligned to mitochondrial RNA, ribosomal RNA, and transfer RNA using Bowtie v1.2.3[71] were discarded as previously described[72]. The remaining reads were aligned to the human genome (hg38) using STAR v2.6[62] and read coverage on known coding genes was calculated using Feature Counts from the Subread package v1.63[63].

Ribotaper[13] was used to detect genomic regions with three-nucleotide periodicity of the dataset and annotate novel sORFs based on Ribo-seq and RNA-seq reads. Ribo-seq alignment files ($n = 24$) were merged together as Ribo_seq.bam and RNA-seq alignment files ($n = 24$) as RNA_seq.bam. Read lengths used were 28, 29, and 30 with an offset of 12 nucleotides for each. Genome annotations were generated using the GRCh38 assembly with the following command: /script_directory/create_annotations_files.bash GRCh38.gtf GRCg38.fasta false false annotation_dir/. After generating the annotation files, SEP calling was carried out as follows: /script_directory/Ribotaper.sh Ribo_seq.bam RNA_seq.bam annotation_dir 28,29,30 12,12,12. The results for CCDS regions were used for downstream analysis.

**Selection pipeline for *i*-Mito-SEPs.** To identify bona fide SEPs, the proteins detected by RiboTaper[13] were shortlisted based on the values provided by Ribo-Taper. Thresholds were applied on the percentage of P-sites in frame, the *p* value of the protein based on the RIBO-profile, the frequency and the ORFscore. Those thresholds were set so that 90% of the 732 SEPs annotated on Uniprot[73] passed every criterion. To identify SEPs of importance in inflammation, only significant differentially expressed genes based on the RNA-seq were considered (*p* adjusted <0.05 and absolute value of the log2FC > 0.5 by DESeq2). Proteins overlapping by more than 50% with the coding sequence of a non-SEP gene were removed. This was done using the Ensembl human GTF 38.95 file[74] and R package tidyverse[75]. Candidates were then further filtered for an overlap between mitochondrial targeting motif prediction and bioinformatic mitochondrial location prediction based on gene signature.

Mitochondrial targeting motifs were predicted in peptides based on a procedure described previously[10]. Briefly, TargetP[76] (mitochondria, Reliability class 1 & 2) and Mitofates[77] (score >0.8) were used to identify SEPs with classical mitochondrial targeting sequences. Transmembrane domains (TMDs) were predicted by TMHMM v2.0[76] or SignalP (4.1 and 5)[78]. Maximum TMD hydrophobicity, as measured by Kyte and Doolittle (1982) was set at 3[79], as we previously noticed that mitochondrial TMDs were on average less hydrophobic compared to non-mitochondrial TMDs[80]. Peptides with twin cysteine pairs (two C9XC, two C3XC, or one C9XC and one C10XC, where X refers to any residues in between two cysteine residues) were also included, as it is characteristic of proteins residing in mitochondria intermembrane.

The mitochondrial location prediction was adapted from our previous clustering method[10]. Four datasets were selected for clustering: 147 samples of immune cells stimulated by LPS from CAD patients (GSE9820)[81], 84 skeletal muscle samples from untrained and trained donors before and after exercise (GSE120862)[82], 82 samples from psoriasis patients (GSE14905)[83], and 202 epithelial colon biopsies from healthy and ulcerative colon patients (GSE26155)[84]. Expression data for genes from the Human MitoCarta[85] as well as 1000 random

genes and the 240 candidate genes were used to collect their signature on more than 6000 gene sets. This was done based on the GeneBridge method[86], where all the selected genes underwent the CAMERA algorithm[87] using limma package[88]. The genes were then clustered on their PCA by the shared nearest neighbor clustering method. For visualization, the UMAP was plotted and the percentage of mitochondrial genes relative to the random genes was calculated for each cluster. Candidates were shortlisted if they were in the top two clusters of at least two datasets among the four. The clustering and UMAP were computed by the Seurat v3.1 package[89].

**Module detection**. In order to keep only the most variable genes, genes with a log2FC above a certain threshold and a corresponding adjusted *p* value below 0.05 in any DESeq2 analysis performed were kept. The threshold on the log2FC was set to 0.5 for the HAECs stimulated with IL-1β RNA-seq, while the threshold for the HAECs infected by Dengue and Zika virus was set to 1. Genes were scaled to be centered on 0 with a unit variance. When approximation of the scaled matrix to a scale-free network by soft-thresholding was possible, the modules were obtained by hierarchical clustering on the computed topology overlap matrix of dissimilarity using R packages WGCNA[90] and DynamicTreeCut[91]. When the approximation was not possible, the hierarchical clustering was directly made on the scaled matrix. The pathway enrichment for each module was made based on the gene ontology biological process database PANTHER[70] or Enrichr[68,69]. Plots were made using library ggplot2[92] and heatmaps using library pheatmap[93].

**Gene module association determination (G-MAD)**. Individual scores per tissue were extracted from Systems-Genetics.org compendium[86] by selecting all datasets of corresponding tissues in the human database (artery, blood, bone marrow, heart, liver, lung, skin, colon). Scores of the same pathways were summed up for each tissue. The signatures were chosen based on their highest and lowest cumulative scores across all selected tissues.

**Analysis of single-cell RNA-seq**. Raw counts were extracted from GEO datasets for mouse CAD model during high-fat diet (GSE131776)[94], human PBMC stimulated by interferon beta (GSE96583)[95], and mouse lung infected by influenza (GSE107947)[96]. Analysis and plots were made using the Seurat v3.1 package[89]. Cell types were defined based on the markers provided by the three papers mentioned.

**GSEA of mito-SEPs vs non-mito-SEPs**. The 50 mitochondrial SEPs present in the Human MitoCarta2.0[85] were plotted against 50 random SEPs from the Uniprot database[73]. Gene set enrichment analysis (GSEA)[97] was computed from the ranked list of gene–gene pairwise correlation for each SEP in a heart cardiomyopathy dataset[12]. Scores for metabolism and inflammation pathways from the Hallmark gene sets of the Molecular Signatures Database (MSgDB)[98] were plotted using the pheatmap package[93].

**Periodicity of MOCCI**. The P-sites were determined from the HAECs IL-1B Ribo-seq dataset for the exonic regions of *C15ORF48* using the filtered read lengths (28, 29, 30 nucleotides) and offsets (12, 12, 12 nucleotides). The exonic region was overlapped with the human fasta file to obtain the amino acid sequence of the ORF.

**Immunofluorescence staining**. Cells were fixed with 4% PFA for 30 min, and permeabilized and blocked with 0.1% Triton X/1% BSA/PBS for 30 min. They were then incubated in primary antibodies diluted in 0.1% Tween 20/1% BSA/PBS overnight at 4 °C. The following primary antibodies were used: anti-MOCCI 1:50 (Sigma-Aldrich HPA012943); anti-TOMM20 1:200 (Abcam ab56783). Cells were washed with PBS before incubating with 4 µg/ml of secondary antibodies for 1 h at room temperature (Alexa Fluor 488, 594, Invitrogen A11001; A21207) and counter stained with 1 µg/mL Hoechst 33342. Samples were imaged on an Olympus FV3000 confocal laser scanning microscope.

**Immunoblotting**. Samples were lysed in RIPA buffer containing complete protein inhibitor cocktail (Roche 04693159001) and 1 mM phenylmethylsulfonyl fluorid (PMSF) (Sigma-Aldrich 10837091001). Samples that were probed for phosphory-lated protein had also PhosSTOP™ (Sigma-Aldrich 4906845001) in the lysis buffer. Protein concentration was measured by bicinchoninic acid (BCA) assay (Ther-moFisher Scientific 23225). Lysates were heated for 5 min in 1X laemmli sample buffer (50 mM Tris-HCl pH 6.8, 2% SDS, 10% glycerol, 12.5 mM EDTA, 0.02% bromophenol blue, 50 mM DTT) at 95 °C. Proteins were then loaded onto 4–12% gradient gels (ThermoFisher NW04122BOX) and transferred to a 0.2 µM PVDF membrane (ThermoFisher 22860) before blocking in 5% non-fat milk TBS-Tween 20 0.1% (v/v) or 5% BSA TBST for 1 h at room temperature. Membranes were probed with primary antibodies in TBST/milk or TBST/BSA overnight at 4 °C. The antibodies used and their dilution factor are as follow: anti-FLAG (Proteintech 20543-I-AP, 1:1000), anti-MTCO-1 (Abcam ab14705; 1:5000), anti-MOCCI (Novus Biologicals NBP1-98391; 1:500), anti-HADHA (ABclonal A13310; 1:5000), anti-GAPDH (Cell Signaling Technology 2118; 1:5000), anti-Histone H3 (Cell Signaling Technology 4499; 1:10000), anti-NDUFA4 (Bioworld Technology BS3883; 1:1000), anti-TOMM70 (ABclonal A4349; 1:5000), anti-TIM23

(Proteintech 11123-1-AP; 1:5000), anti-citrate synthase (Santa Cruz Biotechnology SC-390693; 1:5000), anti-COX4 (ABclonal A10098; 1:4000), anti-UQCRFS1 (Abcam ab14746; 1:2500), anti-UQCRC1 (ABclonal A3339; 1:3000), anti-ATP5A (Santa Cruz Biotechnology SC-136178; 1:5000), anti-COX5B (Abcam ab110263; 1:1000) anti-SDHA (ABclonal A2594; 1:1000), anti-VDAC1 (ABclonal A0810; 1:3000), anti-NDUFA9 (ThermoFisher Scientific 459100; 1:2000), anti-MDA5 (Cell Signaling Technology 5321; 1:1000), anti-RIG1 (Cell Signaling Technology 4520; 1:1000), anti-phosphorylated-p65 (Cell Signaling Technology 3031; 1:1000). Membranes were washed with TBST and probed with HRP/fluorescent protein conjugated secondary antibodies against mouse, rabbit or rat IgG (Jackson ImmunoResearch 715-035-150; 111-035-003; 712-066-153; 1:4000) and washed again. Chemiluminescent or fluorescent signal was captured by Chemidoc imaging system (Bio-Rad).

**Blue native PAGE**. Fifty micrograms of isolated mitochondrial were solubilised in 1X NativePAGE sample buffer (Life Technologies BN2003), 400 µg digitonin, and 0.5% Coomassie G-250 sample additive (Life Technologies BN2004)[99]. The sam-ples were ran on NativePAGE 3–12% gradient gel (Life Technologies BN1001) in dark blue cathode buffer for 30 min at 150 V and in light blue cathode buffer for 150 min at 250 V. Dark blue cathode buffer consisted of 1X NativePAGE Running buffer and 1X NativePAGE Cathode Buffer Additive (Life Technologies BN2007). Light blue cathode buffer was derived from diluting dark blue cathode ten times with 1X NativePAGE Running buffer. The protein was then transferred onto PVDF membrane and fixed in 8% acetic acid for 5 min. After washing with water and air-drying, the membrane was washed with methanol and then water to remove the Coomassie blue. The membrane was then blocked with 5% milk/TBST and immunoblotted with primary antibodies as per described above.

**Two-dimensional BN-SDS PAGE**. The samples were ran on a BN-PAGE, then sliced with a scapel under a microscope into 18 equal pieces. Each piece was crushed and protein was eluted overnight in 50 mM Tris-HCl, 150 mM sodium chloride, and 0.1 mM EDTA, pH 7.5. The gel pieces were pelleted at 10,000xg for 10 min and the supernatant was heated in loading buffer at 95 °C before it was loaded onto a SDS gel. The samples were then immunoblotted as described above.

**Co-immunoprecipitation of MOCCI**. Mitochondria were isolated from AAV-MOCCI or AAV-GFP transduced hearts 2 months after AAV administration. Five hundred micrograms of isolated mitochondria were solubilized with 1% digitonin in IP buffer (50 mM HEPES-KOH, 150 mM NaCl, 1 mM EDTA, pH 7.4 supple-mented with 1X complete, EDTA-free protease inhibitor cocktail, 1 mM PMSF) with the digitonin: mitochondrial protein ratio normalized to 8 g/g. Lysates were incubated at 4 °C for 1 h before they were spun at 20, 000xg for 20 min at 4 °C to remove the insoluble fraction. The supernatant was incubated with 30 µL of anti-FLAG M2 beads (Sigma-Aldrich A2220) for 3 h at 4 °C to allow antigen binding. The unbound material was washed away with 1 mL of IP buffer containing 0.1% digitonin and repeated four times. After removing the residue of IP buffer, beads were incubated with 30 µL of 0.5 mg/mL 3 X FLAG peptide for 30 min at 4 °C to elute bound proteins. Elutes were analyzed by SDS-PAGE followed by immunoblotting.

To enrich for CIV monomers, 1 mg of digitonin-solubilized mouse heart mitochondria were loaded onto a 5–60% sucrose gradient prepared in sucrose gradient buffer (25 mM Tris-HCl, 60 mM KCl, pH 7.4, 0.1% digitonin) and centrifuged at 182,079 × g in a SW41Ti rotor (Beckman) at 4 °C for 18 h. The CIV monomer enriched layer was dialyzed in a dialysis cassette (ThermoFisher 66205) in 500 mL of the IP buffer to remove sucrose. Dialyzed lysates were supplemented with 0.2% digitonin and proceeded to IP as described above.

**Flow cytometry**. To stain for NDUFA4 and MOCCI, cells were trypsinsed and pelleted at 500 × g for 5 min. They were then fixed in 1X fixative buffer for 30 min and permeabilized in 1X permeabilization buffer for 7 min while spinning at 700 × g (Life Technologies 00-5523-00). The cells were then blocked in 1% BSA/permea-bilization buffer for 30 min, before they were incubated in primary antibodies: anti-MOCCI 1:50 (Sigma-Aldrich HPA012943), anti-NDUFA4 1:100 (Santa Cruz Bio-technology SC-517091). After 15 min, they were washed with permeabilization buffer, before they were stained with secondary antibodies at 1:1000 (Alexa Fluor 488, 594, 647, Invitrogen A11008; A11005; A21235). After 15 min, they were washed again and fixed in 2% PFA. The fluorescence intensity was acquired by LSRFortessa™ (BD Biosciences).

To stain for surface proteins, the cells were trypsinized, pelleted, and blocked in 1% BSA/PBS. The cells were then incubated with Near-IR Live/Dead dye (Life Technologies L10119) and primary antibodies for 30 min on ice. The antibodies used were anti-ICAM-1-PE 1:100 (Santa Cruz Biotechnology sc-107 PE), anti-VCAM-1-PE-Vio770 1:100 (Miltenyi Biotec 130-104-127). After washing with 1% BSA/PBS, the cells were fixed in 2% PFA before their fluorescence intensity was acquired by LSRFortessa™ (BD Biosciences) using the BD FACSDiva 8 software.

The data were analyzed by FlowJo version 10.6.1. The cells were gated as shown in Supplementary Fig. 10. Briefly, debris were gated out using the forward and side scatter. Doublets were then excluded based on the height and width measurements in the forward and side scatter. Cells that incorporated the Near-IR Live/Dead dye

were excluded. Any analysis was done on the remaining cells. To identify MOCCI$^+$ cells, a gate was drawn such that only cells with intensity higher than the top 0.5% of the untreated cells were considered positive.

**DENV/ZIKV infection assay.** DENV2 (strain Eden-2) and ZIKV (strain H/PF/2013) were produced in Aedes albopictus C6/36 mosquito cells (ATCC CRL-1660) in complete RPMI 1640 with 25 mM HEPES and 2% FBS at 28 °C and harvested 6 days after infection. Virus titers were determined by a standard plaque assay using BHK-21 cells (ATCC CCL-10)[100,101].

For infection, 200,000 cells were seeded per well in 12-well plates overnight. The next day, they were incubated with DENV or ZIKV for 2 h in serum-free medium and washed with PBS twice. MOI of 1 and 0.1 were used for HAECs and A549, respectively. They were either harvested immediately for determining the input virus amount or were supplemented with complete media and incubated further for 48 h. For the cytokine array and LDH assay, the media was collected, centrifuged at $500 \times g$ for 5 min to remove cell debris and stored in −80 °C. To collect for RNA, cells were washed once with PBS before they were lysed in lysis buffer for RNA extraction as described above. cDNAs were synthesized using iScript Select cDNA synthesis kit (Bio-Rad) using 5 pmol of primer C69B for DENV and 1162 C for ZIKV. DENV and ZIKV genome copy numbers were determined by quantitative real-time PCR using primer pair C14A and C69B with probe VICD2C38B for DENV[102] and primer pair 1086 and 1162 C with probe 1107 for ZIKV[103] with SsoAdvanced Universal Probes Supermix (Bio-Rad 1725281) in a CFX96 Touch Real-Time PCR Detection System (Bio-Rad). The primer and probe sequences are listed in Supplementary Table 2. Copy numbers were then calculated using standard curves generated from serial dilutions of DNA plasmids containing either DENV or ZIKV genome sequence. Fold increase was calculated by dividing the total genomic copy numbers by the input virus amount. Primers and probes were purchased from integrated DNA Technologies, Inc. (IDT).

**Mitochondria isolation.** Fresh mouse hearts were dissected from euthanized animals after a PBS perfusion and rinsed with ice cold PBS containing 10 mM EDTA. Tissues were minced and homogenized in ice cold IBM1 buffer (67 mM sucrose, 50 mM KCl, 1 mM EDTA, 0.2% fatty acid free BSA, 50 mM Tri-HCl, pH 7.4, 1 mM PMSF, and 1X Roche protease inhibitor cocktail) in a Dounce homogenizer. Ten strokes were performed with a motorized pestle that operated at 1300 rpm. The tissue homogenate was spun at $600 \times g$ for 10 min at 4 °C and the supernatant was collected. The supernatant was centrifuged at $1000 \times g$ to remove nuclei, which was kept as a negative control in some experiments. The supernatant was further spun for 15 min at $7000 \times g$, and the supernatant was kept as cytoplasmic fraction. The mitochondrial pellet was resuspended in a buffer consisting of 320 mM Sucrose, 1 mM EDTA, 10 mM Tris-Cl, pH 7.4. The quantity of mitochondrial protein was determined by a BCA assay.

**Mitochondrial membrane fractionation.** Isolated mitochondria were resuspended in isotonic buffer (250 mM sucrose, 1 mM EDTA, 10 mM HEPES-KOH pH 7.4, protease inhibitor cocktail, and 1 mM PMSF) and split into different concentrations of digitonin. After incubating for 1 h at 4 °C, soluble and insoluble fractions were collected by centrifugation at 2000xg for 20 min. The insoluble fraction was resuspended in equal volume of isotonic buffer. Both fractions were analyzed by SDS-PAGE and immunoblotting.

**Protease sensitivity assay.** Isolated mitochondria were resuspended in isotonic buffer (250 mM sucrose, 1 mM EDTA, 10 mM HEPES-KOH pH 7.4) and differentially solubilised in a range of digitonin concentrations or 1% Triton X for 10 min on ice. They were then digested in 100 μg/ml PK (Thermo Scientific EO0491) on ice for 30 min before the digestion was stopped in 8 mM PMSF. Samples were analyzed by SDS-PAGE and immunoblotting.

**Quantitative mass spectrometry.** Isolated heart mitochondria from three individual mice injected with AAV-mMOCCI or AAV-GFP were solubilized in 1% (w/v) sodium deoxycholate and 100 mM HEPES pH 8.1, sonicated for 30 min in water bath sonicator and normalized using the Pierce Protein Assay Kit (ThermoFisher Scientific 23227). Protein pellets (20 μg) were reduced and alkylated with 40 mM chloroacetamide (Sigma 22790) and 10 mM tris (2-carboxyethyl) phosphine hydrochloride (TCEP; BondBreaker, ThermoFisher Scientific 77720) for 5 min at 99 °C with 1500 rpm shaking. Proteins were digested with trypsin (ThermoFisher Scientific) at a 1:50 trypsin:protein ratio overnight at 37 °C. Peptides were labelled with 6plex Tandem Mass Tags (TMT) (ThermoFisher Scientific 90061) in 8:1 label: protein ratio as per the manufacturer instructions. Isopropanol 99% (v/v) and 1% (v/v) trifluoroacetic acid (TFA) was added to the supernatant was transferred to PreOmics cartridges (PreOmics GmbH) before centrifugation at 3000xg at room temperature. Cartridges were washed first with isopropanol (99%) and TFA (1%) solution and then subjected to an additional wash containing 0.2% (v/v) TFA. Peptides were eluted in 80% (v/v) acetonitrile (ACN) and 1% (w/v) ammonium hydroxide, and then acidified to a final concentration of 1% TFA prior to drying in a CentriVap Benchtop Vacuum Concentrator (Labconco).

Samples were fractionated using the Pierce High pH Reversed-Phase Peptide Fractionation Kit (ThermoFisher Scientific 84868) as per the manufacturer's instructions with additional fractions containing 14% and 16% ACN to a total of

ten fractions. Individual fractions were dried using a CentriVap Benchtop Vacuum Concentrator (Labconco) and reconstituted in 0.1% TFA and 2% ACN for mass spectrometry. Liquid chromatography (LC) coupled MS/MS was carried out on an Orbitrap Eclipse Tribrid mass spectrometer (ThermoFisher Scientific) with a nanoESI interface in conjunction with an Ultimate 3000 RSLC nanoHPLC (Dionex Ultimate 3000). The LC system was equipped with an Acclaim Pepmap nano-trap column (Dionex-C18, 100 Å, 75 μm x 2 cm) and an Acclaim Pepmap RSLC analytical column (Dionex-C18, 100 Å, 75 μm x 50 cm). The tryptic peptides were injected into the trap column at an isocratic flow of 5 μL/min of 0.1% (v/v) formic acid for 5 min applied before the trap column was switched in line with the analytical column. The eluents were 5% DMSO in 0.1% v/v formic acid (solvent A) and 5% DMSO in 100% v/v CH3CN and 0.1% v/v formic acid (solvent B). The flow gradient was (i) 0–6 min at 0% B, (ii) 6–65 min, 3–23% B (iii) 65–75 min 23–40% B (iv) 75–80 min, 40–80% B (v) 80–85 min, 80–80% B (vi) 85–85.1 min, 80–3% and equilibrated at 3% B for 10 min before the next sample injection. The mass spectrometer was operated in positive-ionization mode with spray voltage set at 1.9 kV and source temperature at 275 °C. The mass spectrometer was operated in the data-dependent acquisition mode MS spectra with APD mode on and scanning from m/z 375-1500 at 120k resolution with AGC target of 4e5. The "top speed" acquisition method mode (3 sec cycle time) on the most intense precursor was used whereby peptide ions with charge states ≥2–7 were isolated with isolation window of 1.6 m/z and fragmented with high energy collision (HCD) mode with stepped collision energy of 35 ± 5%. Fragment ion spectra were acquired in Orbitrap at 15k resolution. Dynamic exclusion was activated for 30 s.

Raw files were processed using the MaxQuant platform (version 1.6.10.43)[104] and searched against UniProt mouse database with canonical and isoforms (November 2019) using default settings for a TMT 6plex experiment with applied correction values for TMT isotopes and "label min. ratio count" set to "1". The proteinGroups.txt output from the search was processed in Perseus (version 1.6.10.43)[105]. Briefly, log2-transformed TMT reporter intensity corrected values were grouped into three technical replicates of mitochondria isolated from AVV-mMocci or AAV-GFP. Positive entries for "potential contaminant", "reverse", and "only identified by site" were removed from the dataset and normalized by subtraction of known mitochondrial entries from mouse known and predicted IMPI database (2017) matched by gene name. A two-tailed Student's $t$ test was performed for MOCCI vs GFP with significance determined by $p$ value ≤0.05 and volcano plots generated using scatter plot function in Perseus.

**Cellular respirometry and mitochondria flux assays.** For the Mito Stress Test, Seahorse XF96 Cell Culture Microplates (Agilent Technologies 101085-004) were coated with poly-L-lysine and 18,000 cells were seeded into each well. After 24 h, the media was replaced with XF basal DMEM supplemented with 1 mM pyruvate (Sigma-Aldrich S8636), 2 mM glutamine (Life Technologies 25030081), and 10 mM glucose (Sigma-Aldrich G8644) and incubated at 37 °C with no CO$_2$ for 45 min. XFe96 sensor cartridges (Agilent) were hydrated in water the night before and calibrated with Seahorse XF Calibrant (Agilent) on the assay day at 37 °C with no CO$_2$. For the assay itself, oligomycin (Sigma-Aldrich G8644), FCCP (Sigma-Aldrich C2920), and Rotenone/Antimycin A (Sigma R8875; A8674) were injected sequentially to reach final concentrations of 2 μM, 1 μM, and 1 μM, respectively. The data was collected via the Seahorse Wave, Agilent software. BCA assay (ThermoFisher Scientific 23225) was used to quantify the amount of protein in each well for normalization.

For the mitochondrial flux assays, mitochondria were isolated from the hearts of AAV-injected mice and plated at 1.0–1.4 μg/well of a Seahorse XF96 Cell Culture Microplate. The assay buffer contained 5 mM pyruvate, 1 mM malate, 70 mM sucrose, 220 mM mannitol, 5 mM potassium phosphate monobasic, 5 mM magnesium chloride, 2 mM HEPES, 1 mM EGTA, and 0.2% fatty acid free BSA[106]. For CIV activity, 4 μM FCCP was included in the assay buffer. Antimycin A, N,N, N',N'-Tetramethyl-p-phenylenediamine dihydrochloride (TMPD; Sigma-Aldrich T7394) and potassium azide (Sigma-Aldrich 740411) were injected sequentially to final concentrations of 1 μM, 100 μM, 15 mM, respectively. For the coupled assay, ADP, oligomycin A, FCCP and Antimycin A were injected sequentially to final concentrations of 1 mM, 2 μM, 1 μM, and 1 μM, respectively. The amount of mitochondria loaded in each well was normalized by citrate synthase activity. The amount of respiration due to CIV activity is quantified by the difference between the maximal rate after TMPD injection and the minimal rate after potassium azide injection.

To measure the CIV or CI activity in cells, 18,000 cells per well were seeded onto poly-L-lysine coated Seahorse XF96 Cell Culture Microplates (Agilent Technologies 101085-004). The cells were switched to MAS Buffer (70 mM sucrose, 220 mM mannitol, 10 mM potassium dihydrogen phosphate, 5 mM magnesium chloride, 2 mM HEPES, 1 mM EGTA, adjusted to pH 7.2) containing 0.4% fatty acid free BSA right before the assay. For CIV activity, in the first injection, TMPD (Sigma-Aldrich T7394), ascorbate (Sigma-Aldrich A5960), ADP (Sigma-Aldrich A5285), FCCP (Sigma-Aldrich C2920), and digitonin were injected to reach final concentrations of 0.5 mM, 2 mM, 1 mM, 1 μM, and 0.0025%, respectively. The second and third injections contained oligomycin and potassium azide to achieve final concentrations of 2 μM and 30 mM, respectively. For CI activity, sodium pyruvate, malate, ADP, FCCP, and digitonin were injected to reach final concentrations of 5 mM, 2.5 mM, 1 mM, 1 μM, and 0.0025%, respectively. The

second and third injections contained oligomycin and rotenone to achieve final concentrations of 2 μM and 1 μM, respectively. The number of cells were normalized either by CS activity using the same plate or BCA on a separate plate.

To quantify citrate synthase activity, the buffer was removed and 113 μl of CS buffer (200 mM Tris buffer at pH 8.0, 0.2% Triton X-100 (v/v), 10 μM DTNB (Sigma D8130), 1 mM Acetyl-CoA (Sigma-Aldrich A2181)) was added per well. Five microliters of 10 mM oxaloacetate (Sigma O4126) was added to each well. The plate was immediately placed into the Tecan Microplate Reader M200 (Tecan). Absorbance at 412 nm at 37 °C was recorded with at minimal time interval for 8 min and citrate synthase activity was then calculated using the formula below

$$CS\,activity = \sum_{i=1}^{n} \left[ \left( \frac{A_i - A_0}{t_i - t_0} \right) \right] / n \left( n \in N^* \right)$$

$n$ = total number of absorbance records; $A$ = absorbance recorded; $t$ = time.

**Respiratory chain enzymatic assay**. About 18,000 cells per well were seeded onto poly-L-lysine coated Seahorse XF96 Cell Culture Microplates (Agilent Technologies 101085-004). After treatment, the media was removed and the plate washed once with PBS. 50 microliters of 50 mM potassium phosphate, 62.5 μM cytochrome c, and 0.002% digitonin, with the pH adjusted to 7.4 was added per well. The plate was measured at 550 nm for 15 min in the Tecan Microplate Reader M200 (Tecan). Final concentration of 30 mM potassium cyanide was then added to each well and the plate was measured at 550 nm for another 15 min. The activity of CIV was then inferred from the difference in the rate of change of absorbance between pre- and post- cyanide addition.

**Mitochondria membrane potential and ROS**. The concentrations used for TMRE (ThermoFisher Scientific T669), JC-1 (Life Technologies 65-0851-38), DCF-DA (Sigma-Aldrich 6883), and mitoSOX™ (ThermoFisher Scientific M36008) were 100 nM, 1.5 μM, 10 μM, and 5 μM, respectively. Cells were incubated in EGMV2 media with the dyes for 30 min. The cells were then washed with warm PBS, trypsinized and pelleted, before resuspending in warm media for acquisition by flow cytometry.

For Amplex Red assays, 20,000 A549 cells per well were seeded in 96-well plate. The next day, the cells were washed twice with PBS and incubated with 60 μL of PBS for 30 min. 50 microliters of the PBS was then incubated with the Amplex Red reagent/HRP working solution as per the manufacturer's instructions for 30 min and acquired on Tecan Infinite M200.

**Cytokine array**. Media that were conditioned by endothelial cells treated with LPS or virus were collected after 24–48 h as stated and centrifuged at 500xg to remove cell debris. Cytokines from the supernatant were quantified using LEGENDplex™ (BioLegend 740809; 740503) according to the manufacturer's instructions. For DENV/ZIKV and LPS treated samples, the media were diluted 5X by the assay buffer before adding them to the LEGENDplex beads. Fluorescence intensity of the beads were acquired by LSRFortessa™ (BD Biosciences). The levels of cytokines were analyzed using the LEGENDplex™ v8.0 software. BCA assay (ThermoFisher 23225) was used to normalize cell number variation.

**Lactate dehydrogenase assay**. Ten microliters of conditioned media was transferred from each condition into 384-well plates (Merck 781098). 0, 2.5, 5, 7.5, 10, and 12.5 nmole/well of NADH standards were prepared as suggested by the kit (Sigma-Aldrich MAK066). After the addition of 10 μL LDH substrate into each well, the plate was shook for 60 s before the plate was read using a Tecan Infinite M200 at 37 °C every 1 min interval. The concentration of LDH was calculated based on the manufacturer's protocol.

**Treatment with ROS scavenger**. About 10,000 cells/well were seeded into 96-well plates. The next day, the cells were treated with 1 mM NAC, pH 7.4 (Sigma-Aldrich A9165). Fresh media were added to untreated cells. After 24 h, half of the wells were treated with 1 μg/mL LPS while still containing the scavengers. After 16 h, the media was collected, pelleted to remove debris, and stored at −80 °C. To ensure that the ROS levels were reduced in these cells, the ROS levels were checked by DCF. About 100,000 cells/well were seeded in 24-well plates at the same time and treated with the NAC for 48 h. They were stained with DCF and analyzed as described above.

**NF-κB activity assay**. Lentivirus was produced from the transfer plasmid pHAGE NFkB-TA-LUC-UBC-GFP-W (Addgene 49343; a gift from Darrell Kotton)[107]. Endothelial cells were transduced with the NF-κB reporter virus and checked for successful transduction using the GFP reporter in the vector. After a week post-transduction, the cells were seeded at 2000 cells per well in 384-well plates (Merck 781098) in 15 μL media. The next day, they were treated with 1 ng/mL IL1b for the indicated length of time, bringing the final volume in each well to 20 μL. 20 microliters of Bright-Glo™ Luciferase Assay reagent (Promega E2610) was added to each well and acquired using a Tecan Infinite M200. Each experiment was run in quadruplicates.

**Protein structure prediction**. Predicted model of MOCCI structure was derived using iTASSER[108–110]. The highest scoring MOCCI model was then docked with that of NDUFA4 (5Z62 chain N) in Maestro (Schrodinger Suites).

**Protein alignment**. The protein sequences were aligned with Clustal Omega[111]. The output was then piped into Jalview[112] for clustering and visualization.

**Quantification and statistical analysis**. The statistical methods used are listed in the captions and the $p$ values are written on the figures. The graphs and statistical tests were generated with GraphPad Prism 8.

**Reporting Summary**. Further information on research design is available in the Nature Research Reporting Summary linked to this article.

## Data availability

The RNA-seq data have been deposited in Sequence Read Archive under the BioProject ID PRJNA672723. The code to identify mito-SEPs are uploaded in GitHub [https://github.com/LenaHoLab/Lee-et-al-2021-R-code]. The mass spectrometry proteomics data have been deposited to the ProteomeXchange Consortium via the PRIDE[113] partner repository with the dataset identifier PXD024438. Source data are provided with this paper.

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

## Acknowledgements

We thank Srikanth Nama for assistance with lentiviral production, Shiqi Lim, Anissa Widjaja, and Eleonora Adami for their technical guidance in Ribo-seq library preparation, Sheryl Beh, Peh Jih Hou, and Liang Chao for technical assistance in RNA-seq, molecular biology, and biochemistry. We thank the Duke-NUS animal facility, FACs facility, the A*STAR/RSC microscopy platform (Graham Wright), and the Bio21 Mass Spectrometry and Proteomics Facility (MMSPF) for their enabling technologies and services. D.A.S. is supported by Australian National Health & Medical Research Council (NHMRC) grants GNT1140851 and GNT1140906. C.Q.E.L. is supported by A*STAR SSDF fellowship. W.L.C. and D.L. acknowledge funding from A*STAR IAF-PP (H17/01/a0/012), V.T. acknowledges support from NRF-CRP17-2017-02. A.L.S. acknowledges funding from NRF-CRP17-2017-04 and thanks the European Virus Archive for strain H/PF/2013. This work is funded by fellowships NRF-NRFF2017-05 (National Research Foundation of Singapore) and HHMI-IRSP55008732 (Howard Hughes Medical Institute International Research Scholar Program) awarded to L.H.

## Author contributions

L.H. and C.Q.E.L. conceptualized the study, designed all experiments, and wrote the manuscript. C.Q.E.L. performed all experiments with help from R.N. and Y.C. and V.T. H., B.K., and S.C. performed all bioinformatics analyses with supervision from S.S. and O.J.L.R. S.Z. performed mitochondrial assays and biochemistry. D.A.S. and D.H.H. carried out proteomic analyses. C.K.M. and A.L.S. performed DENV/ZIKV infections and viral genome quantifications. D.L. and W.L.C. provided AAVs and R.L. administered them to mice. F.L.Z. provided CRISPR constructs and V.T. provided expertize in inflammation signaling.

## Competing interests

The authors declare no competing interests.
