## [Peer Review File · Nature Communications]

REVIEWERS' COMMENTS

Reviewer #1 (Remarks to the Author):

The authors have made an attempt to address a number of concerns raised following the initial submission of this manuscript and while some of the newly provided, additional data/discussion included in this revised manuscript are helpful, I feel some issues remain.

Most notably, the link between ROS, membrane potential and inflammation is still poorly explored. The authors agree that ROS reduction is unlikely to play a role in effects on cytokine production and instead propose that complex IV activity reduction is responsible for the decrease in cytokine production. What is the mechanism of this? The fact that WT-mRNA reversed the reduction in ROS by MOCCI is very perplexing.

Furthermore, the data in the revised manuscript (R Fig. 4) is an inappropriate assessment of the effects of mitoQ in their system. MitoQ is a mitochondrial ROS scavenger, DCF is not specific reagent for the assessment of mitoROS. Given these issues I feel the data on ROS is lacking and should not be proposed as a mechanism for MOCCI-associated decreases in inflammation.

What does the data in the new Figure 7a (left) depict? It is unclear to me what these violin plots show.

Overall, I think the author's model (depicted in Figure. 8h) needs be readjusted. While these data support a role for MOCCI and miR-147b in decreasing CIV activity, and in limiting some inflammatory phenotypes, the link between these effects is unclear and either needs to be further explored experimentally/stated less strongly in the text.

Reviewer #3 (Remarks to the Author):

The authors have addressed all my concerns in the revised manuscript. I think that additional experiments have strengthened their claims.

Reviewer #1 (Remarks to the Author):

The authors have made an attempt to address a number of concerns raised following the initial submission of this manuscript and while some of the newly provided, additional data/discussion included in this revised manuscript are helpful, I feel some issues remain. Most notably, the link between ROS, membrane potential and inflammation is still poorly explored. The authors agree that ROS reduction is unlikely to play a role in effects on cytokine production and instead propose that complex IV activity reduction is responsible for the decrease in cytokine production. What is the mechanism of this? The fact that WT-mRNA reversed the reduction in ROS by MOCCI is very perplexing.

The mechanism of how CIV activity affects cytokine production is an important question that is clinically relevant. Interestingly, Karan et al (2020) treated human PBMCs with inhibitors against the different complexes within the ETC and found that inhibiting CIV has the most robust reduction of pro-inflammatory cytokines. Searching for the mechanism will be the focus of a follow-up paper. Since we do not provide a mechanism, we have inserted “?” in the arrow linking CIV activity reduction with reduction in cell death and pro-inflammatory cytokine production in Fig. 8h (see below).

We agree with the reviewer that the reversal of ROS reduction in WT-mRNA is intriguing. As explained in line 297-301 (line 294 in original manuscript), the reversal cannot be explained by the additional presence of miR-147b alone, as transfection with the miR-147b in MOCCI cells did not revert the ROS levels. It is not also due to the artificial overexpression of the MOCCI protein, since the WT-mRNA cells also have the MOCCI protein. We therefore alluded to the possibility that the WT-mRNA transcript has additional (long) non-coding functions that were not explored in this manuscript. All in all, the modulation of ROS by the *C15ORF48* gene is multifaceted and deserves further investigation, which will be the focus of future work.

To reduce the emphasis on the ROS as a mechanism, we have altered line 496-502 (line 496 in original manuscript) to the following:

“How CIV activity controls cytokine production is an open area of investigation. One possible mechanism, as seen with MOCCI expression, is the reduction of ROS production, but that does not appear to be the case since miR-147b suppresses cytokine production without reducing ROS. CIV inhibition could modulate cytokine production indirectly through retrograde signaling pathways that are activated during mitochondrial stress³⁹. A growing consensus points to mitochondrial retrograde signaling as a type of integrated stress response^{40,41}. An intriguing possibility is that changes in CIV activity are sensed and relayed to downstream effectors through similar mechanisms.”

Furthermore, the data in the revised manuscript (R Fig. 4) is an inappropriate assessment of the effects of mitoQ in their system. MitoQ is a mitochondrial ROS scavenger, DCF is not specific reagent for the assessment of mitoROS. Given these issues I feel the data on ROS is lacking and should not be proposed as a mechanism for MOCCI-associated decreases in inflammation.

Our experiments with Mito-SOX yielded similar results as DCF, prompting us to investigate if MitoQ, like NAC, is sufficient to mimic the effects of MOCCI. We have specifically mentioned in the text at line 357-359 (line 351-354 in the original manuscript) and line 496-

498 (line 493-495 in the original manuscript) that we do not think that ROS can account for the decreased inflammation observed, albeit a bona fide functional consequence of MOCCI expression.

What does the data in the new Figure 7a (left) depict? It is unclear to me what these violin plots show.

Thank you for pointing out this ambiguity. Genes were clustered according to their expression pattern after treatment with IL1b for 45min, 12h and 24h. Genes with similar expression pattern were assigned into the same module. The violin plots show the gene expression pattern of the four modules with the most number of genes. To make this information clearer, we have annotated on the x-axis the time point at which the violin plot was derived from. We have also given more explanation in the figure legend.

Line 1606 (line 906 in the original manuscript): “7a. (Left) Differentially expressed genes in the HAECs treated with IL-1 β across the timepoint series (Fig. 1b) were clustered into co-expression modules by dynamic tree cut. **The log fold change is relative to the untreated cells.**”

Left (Original plot); Right (Modified plot)

Overall, I think the author’s model (depicted in Figure. 8h) needs be readjusted. While these data support a role for MOCCI and miR-147b in decreasing CIV activity, and in limiting some inflammatory phenotypes, the link between these effects is unclear and either needs to be further explored experimentally/stated less strongly in the text.

We have modified Figure 9c (Original Fig 8h) with question marks to emphasise that the exact mechanism linking CIV inhibition by MOCCI and reduced inflammatory outcomes is not known yet. We have also included the following in the figure legend: “The exact mechanism on how inhibiting CIV activity can modulate cytokine secretion and cell death remains an outstanding question.”

Reviewer #3 (Remarks to the Author):

The authors have addressed all my concerns in the revised manuscript. I think that additional experiments have strengthened their claims.

We thank the Reviewer for their previous feedback which strengthened this paper significantly.

Other things that were changed:

- Line 253. Inserted the bold words to be more specific: “Therefore, **in vitro cultured human** monocytes and macrophages produce only miR-147b from the C15ORF48 locus.”
- Line 569-572. Removed the strikethrough words.

“For instance, in human PBMCs, C15ORF48 mRNA is expressed constitutively at high levels in monocytes (Fig. S2a). Results from our THP-1 model in vitro predict that these monocytes express high levels of miR-147b ~~without MOCCI translation at basal~~ (Fig. S5a-d), consistent with their main function of immune surveillance and viral defense.”

- Added the following at lines 364-367 to cite a recent published paper:

“Intriguingly, MOCCI has a viral homologue in poxviruses, suggesting that it has been evolutionarily selected and co-opted by viruses for immune evasion via modulation of the interferon response.”